# Finding Local Minima Efficiently in Decentralized Optimization

**Wenhan Xian**
Computer Science
University of Maryland College Park
`wxian1@umd.edu`

**Heng Huang** *
Computer Science
University of Maryland College Park
`heng@umd.edu`

## Abstract

In this paper we study the second-order optimality of decentralized stochastic algorithm that escapes saddle point efficiently for nonconvex optimization problems. We propose a new pure gradient-based decentralized stochastic algorithm PEDESTAL with a novel convergence analysis framework to address the technical challenges unique to the decentralized stochastic setting. Our method is the first decentralized stochastic algorithm to achieve second-order optimality with non-asymptotic analysis. We provide theoretical guarantees with the gradient complexity of $\tilde{O}(\epsilon^{-3})$ to find $O(\epsilon, \sqrt{\epsilon})$-second-order stationary point, which matches state-of-the-art results of centralized counterparts or decentralized methods to find first-order stationary point. We also conduct two decentralized tasks in our experiments, a matrix sensing task with synthetic data and a matrix factorization task with a real-world dataset to validate the performance of our method.

## 1   Introduction

Decentralized optimization is a class of distributed optimization that trains models in parallel across multiple worker nodes over a decentralized communication network. Decentralized optimization has recently attracted increased attention in machine learning and emerged as a promising framework to solve large-scale tasks because of its capability to reduce communication costs. In the conventional centralized paradigm, all worker nodes need to communicate with the central node, which results in high communication cost on the central node when the number of nodes is large or the transmission between the center and some remote nodes suffers network latency. Conversely, decentralized optimization avoids these issues since each worker node only communicates with its neighbors.

Although decentralized optimization has shown advantageous performance in many previous works (Lian et al. [2017], Tang et al. [2018]), the study of second-order optimality for decentralized stochastic optimization algorithms is still limited. Escaping saddle point and finding local minima is a core problem in nonconvex optimization since saddle point is a category of first-order stationary point that can be reached by many gradient-based optimizers such as gradient descent but it is not the expected point to minimize the objective function.

Perturbed gradient descent (Jin et al. [2017]) and negative curvature descent (Xu et al. [2018], Allen-Zhu and Li [2018]) are two primary pure gradient-based methods (not involving second-order derivatives) to achieve second-order optimality. Typically, perturbed gradient descent method is composed of a descent phase and an escaping phase. If the norm of gradient is large, the algorithm will run the descent phase as normal. Otherwise it will run the escaping phase to discriminate whether the candidate first-order stationary point is a saddle point or local minimum. Negative curvature

---

*This work was partially supported by NSF IIS 1838627, 1837956, 1956002, 2211492, CNS 2213701, CCF 2217003, DBI 2225775.

37th Conference on Neural Information Processing Systems (NeurIPS 2023).

descent method escapes saddle point by computing the direction of negative curvature at the candidate point. If it is categorized as a saddle point then the algorithm will update along the direction of negative curvature. Generally it involves a nested loop to perform the negative curvature subroutine.

Currently, the solution to the second-order optimality of decentralized problem in deterministic setting has been proposed. Perturbed Decentralized Gradient Tracking (PDGT) (Tziotis et al. [2020]) is a decentralized deterministic algorithm adopting the perturbed gradient descent strategy to achieve second-order stationary point. However, it is expensive to compute full gradients for large machine learning models. It is crucial to propose a stochastic algorithm to obtain second-order optimality for decentralized problems. Besides, there are some drawbacks of PDGT to make it less efficient and hard to be generalized to the stochastic setting. These drawbacks are also the key challenges to achieve second-order optimality for decentralized algorithms, which are listed as follows:

(1) PDGT runs fixed numbers of iterations in descent phase and escaping phase such that the phases of all nodes can be changed simultaneously. This strategy works because the descent is easy to be estimated in deterministic setting. Nonetheless, the exact descent of stochastic algorithm over a fixed number of iterations is hard to be bounded because of randomness and noises. If the fixed number is not large enough it is possible that the averaged model parameter is not a first-order stationary point. If the fixed number is as large as the expected number of iterations to achieve first-order stationary point, the algorithm will become less efficient as it is probably stuck at a saddle point for a long time before drawing the perturbation, especially in the second and later descent phase. Specifically, applying fixed number of iterations in each phase results in the complexity of at least $\tilde{O}(\epsilon^{-4.5})$ (see Appendix D), which is higher than $\tilde{O}(\epsilon^{-3})$ of our method. Therefore, we are motivated to propose an algorithm that can change phases *adaptively* (based on runtime gradient norm) and *independently* (not required to consider status on other nodes or notify other nodes).

(2) In PDGT the perturbations on all nodes are drawn from the same random seed. Besides, a coordinating protocol involving broadcast and aggregation is used to compute the averaged model parameter and the descent of overall loss function to discriminate the candidate point. These strategies together with the fixed number of iterations act as a hidden coordinator to make PDGT discriminate saddle point in the same way as centralized algorithms. However, when the number of worker nodes is large it is time-consuming to perform broadcast or aggregation over the whole decentralized network. Moreover, when generalized to stochastic setting the changing of phase is not guaranteed to be synchronized. Additionally, we will note in the Supplementary Material that the consensus error $\frac{1}{n}\sum_{i=1}^{n}\|x_t^{(i)} - \bar{x}_t\|^2$ is another factor to impact the effectiveness of perturbed gradient descent, which is not present in centralized problems. All above issues are theoretical difficulties to study and ensure second-order optimality for decentralized stochastic algorithms.

(Vlaski and Sayed [2020]) proves the theoretical guarantee of second-order optimality for decentralized stochastic algorithm with perturbed gradient descent. However, it does not provide a non-asymptotic analysis to estimate the convergence rate or gradient complexity. The effectiveness of the result relies on a sufficiently small learning rate, and it does not present a specific algorithm. The analysis is based on the assumption that the iteration formula can be approximated by a centralized update scheme when the learning rate is small enough. Nevertheless, in practice it is difficult to maintain an ideally small learning rate, and the iterative update process can be more complex as previously mentioned. To our best knowledge, the second-order optimality issue of decentralized stochastic algorithm with non-asymptotic analysis is still not solved. Therefore, we are motivated to study this important and challenging issue and raise the following questions:

*Can we design a decentralized stochastic optimization algorithm with non-asymptotic analysis to find local minima efficiently? Is the algorithm still effective to discriminate saddle point even if each node can change its phase adaptively and independently without any coordinating protocols?*

The answer is affirmative. In this paper, we propose a novel gradient-based algorithm named PErturbed DEcentralized STORM ALgorithm (PEDESTAL) which is the first decentralized stochastic algorithm to find second-order stationary point. We adopt perturbed gradient descent to ensure the second-order optimality and use STORM (Cutkosky and Orabona [2019]) estimator to accelerate the convergence. We provide completed convergence analysis to guarantee the second-order optimality theoretically. More details about the reason of choosing perturbed gradient descent and technical difficulties are discussed in Section 3.2. Next we will introduce the problem setup in this paper.

We focus on the following decentralized optimization problem:

$$\min_x f(x) = \frac{1}{n} \sum_{i=1}^n f_i(x), \; f_i(x) = \mathbb{E}_{\xi \sim D_i} F_i(x, \xi) \tag{1}$$

where $n$ is the number of worker nodes in the decentralized network and $f_i$ is the local loss function on $i$-th worker node. Here $f_i$ is supposed to take the form of stochastic expectation over local data distribution $D_i$, which covers a variety of optimization problems including finite-sum problem and online problem. Data distributions on different nodes are allowed to be heterogeneous. The objective function $f$ is nonconvex such that saddle points probably exist.

The goal of our method is to find $O(\epsilon, \epsilon_H)$-second-order stationary point of problem 1, which is defined by the point $x$ satisfying $\|\nabla f(x)\| \leq \epsilon$ and $\min eig(\nabla^2 f(x)) \geq -\epsilon_H$, where $eig(\cdot)$ represents the eigenvalues. The classic setting is $\epsilon_H = \sqrt{\epsilon}$.

We summarize the contributions of this paper as follows:

- We propose a novel algorithm PEDESTAL, which is the first decentralized stochastic gradient-based algorithm to achieve second-order optimality with non-asymptotic analysis.
- We provide a new analysis framework to support changing phases adaptively and independently on each node without any coordinating protocols involving broadcast or aggregation. We also address certain technical difficulties unique to decentralized optimization to justify the effectiveness of perturbed gradient descent in discriminating saddle point.
- We prove that our PEDESTAL achieves the gradient complexity of $\tilde{O}(\epsilon^{-3} + \epsilon\epsilon_H^{-8} + \epsilon^4\epsilon_H^{-11})$ to find $O(\epsilon, \epsilon_H)$-second-order stationary point. Particularly, PEDESTAL achieves the gradient complexity of $\tilde{O}(\epsilon^{-3})$ in the classic setting $\epsilon_H = \sqrt{\epsilon}$, which matches state-of-the-art results of centralized counterparts or decentralized methods to find first-order stationary point.

## 2 Related Work

In this section we will introduce the background of related works. The comparison of important features is shown in Table 1. Here $\tilde{O}(\cdot)$ refers to the big $O$ notation that hides the logarithmic terms.

### 2.1 Decentralized Algorithms for First-Order Optimality

Decentralized optimization is an efficient framework to solve problem 1 collaboratively by multiple worker nodes. In each iteration a worker node only needs to communicate with its neighbors. One of the best-known decentralized stochastic algorithm is D-PSGD (Lian et al. [2017]), which integrates average consensus with local stochastic gradient descent steps and shows competitive result to centralized SGD. The ability to address Non-IID data is a limitation of D-PSGD and some variants of D-PSGD are studied to tackle the data heterogeneity issue, such as $D^2$ (Tang et al. [2018]) by storing previous status and GT-DSGD (Xin et al. [2021b]) by using gradient tracking (Xu et al. [2015], Lorenzo and Scutari [2016]). D-GET (Sun et al. [2020]) and D-SPIDER-SFO (Pan et al. [2020]) improve the gradient complexity of D-PSGD from $O(\epsilon^{-4})$ to $O(\epsilon^{-3})$ by utilizing variance reduced gradient estimator SPIDER (Fang et al. [2018]). GT-HSGD also achieves gradient complexity of $O(\epsilon^{-3})$ by combining gradient tracking and STORM gradient estimator (Cutkosky and Orabona [2019]). SPIDER requires a large batchsize of $O(\epsilon^{-1})$ on average and a mega batchsize of $O(\epsilon^{-2})$ periodically. In contrast, STORM only requires a large batch in the first iteration. After that the batchsize can be as small as $O(1)$, which makes STORM more efficient to be implemented in practice.

### 2.2 Centralized Algorithms for Second-Order Optimality

Perturbed gradient descent is a simple and effective method to escape saddle points and find local minima. PGD (Jin et al. [2017]) is the representative of this family of algorithms, which achieves second-order optimality in deterministic setting. It draws a perturbation when the gradient norm is small. If this point is a saddle point, the loss function value will decrease by a certain threshold within a specified number of iterations (*i.e.*, breaking the escaping phase) with high probability. Otherwise, the candidate point is regarded as a second-order stationary point. In stochastic setting, Perturbed SGD perturbs every iteration and suffers a high gradient complexity of $O(\epsilon^{-8})$ to achieve $O(\epsilon, \sqrt{\epsilon})$-second-order stationary point and the gradient complexity hides a polynomial factor of dimension $d$. CNC-SGD requires a Correlated Negative Curvature assumption and the gradient complexity of $\tilde{O}(\epsilon^{-5})$ to achieve the classic second-order optimality. SSRGD (Li [2019]) adopts the

| Name | Averaged Batchsize | Gradient Complexity | Classic Setting |
|---|---|---|---|
| D-PSGD [12] | $O(1)$ | $O(\epsilon^{-4})$ | - |
| GT-DSGD [22] | $O(1)$ | $O(\epsilon^{-4})$ | - |
| D-GET [16] | $O(\epsilon^{-1})$ | $O(\epsilon^{-3})$ | - |
| D-SPIDER-SFO [15] | $O(\epsilon^{-1})$ | $O(\epsilon^{-3})$ | - |
| GT-HSGD [21] | $O(1)$ | $O(\epsilon^{-3})$ | - |
| SGD+Neon2 [1] | $O(1)$ | $\tilde{O}(\epsilon^{-4}+\epsilon^{-2}\epsilon_H^{-3}+\epsilon_H^{-5})$ | $\tilde{O}(\epsilon^{-4})$ |
| SCSG+Neon2 [1] | $O(\epsilon^{-0.5})$ | $\tilde{O}(\epsilon^{-10/3}+\epsilon^{-2}\epsilon_H^{-3}+\epsilon_H^{-5})$ | $\tilde{O}(\epsilon^{-3.5})$ |
| Natasha2+Neon2 [1] | $O(\epsilon^{-2})$ | $\tilde{O}(\epsilon^{-3.25}+\epsilon^{-3}\epsilon_H^{-1}+\epsilon_H^{-5})$ | $\tilde{O}(\epsilon^{-3.5})$ |
| SPIDER-SFO$^+$ [5] | $O(\epsilon^{-1})$ | $\tilde{O}(\epsilon^{-3}+\epsilon^{-2}\epsilon_H^{-2}+\epsilon_H^{-5})$ | $\tilde{O}(\epsilon^{-3})$ |
| Perturbed SGD [6] | $O(1)$ | $O(\epsilon^{-4}+\epsilon_H^{-16})$ | $O(\epsilon^{-8})$ |
| CNC-SGD [4] | $O(1)$ | $\tilde{O}(\epsilon^{-4}+\epsilon_H^{-10})$ | $\tilde{O}(\epsilon^{-5})$ |
| SSRGD [11] | $O(\epsilon^{-1})$ | $\tilde{O}(\epsilon^{-3}+\epsilon^{-2}\epsilon_H^{-3}+\epsilon^{-1}\epsilon_H^{-4})$ | $\tilde{O}(\epsilon^{-3.5})$ |
| Pullback [2] | $O(\epsilon^{-1})$ | $\tilde{O}(\epsilon^{-3}+\epsilon_H^{-6})$ | $\tilde{O}(\epsilon^{-3})$ |
| PDGT [18] | Full | - | - |
| PEDESTAL-S (ours) | $O(1)$ | $\tilde{O}(\epsilon^{-3}),\ \epsilon_H \geq \epsilon^{0.2}$ | - |
| PEDESTAL (ours) | $O(\epsilon^{-3/4})$ | $\tilde{O}(\epsilon^{-3}+\epsilon\epsilon_H^{-8}+\epsilon^4\epsilon_H^{-11})$ | $\tilde{O}(\epsilon^{-3})$ |

Table 1: The comparison of important properties between related algorithms and our PEDESTAL. Column "Averaged Batchsize" is computed when $\epsilon_H = \sqrt{\epsilon}$. Column "Classic Setting" refers to the gradient complexity under the classic condition $\epsilon_H = \sqrt{\epsilon}$. The first group of algorithms are decentralized methods achieving first-order optimality. The second group of algorithms are centralized methods achieving second-order optimality. The last group of algorithms are decentralized methods achieving second-order optimality. PEDESTAL-S is a special case of PEDESTAL with $O(1)$ batchsize. The complexity of PDGT is not shown because it is not stochastic.

same two-phase scheme as PGD but uses the moving distance as the criterion to discriminate saddle point in the escaping phase. It also takes advantage of variance reduction to improve the gradient complexity to $\tilde{O}(\epsilon^{-3.5})$. Pullback (Chen et al. [2022]) proposes a pullback step to further enhance the gradient complexity to $\tilde{O}(\epsilon^{-3})$, which matches the best result of reaching first-order stationary point.

### 2.3 Stochastic Gradient Descent

A branch of study of stochastic gradient descent argues that SGD can avoid saddle point under certain conditions. However, that is completely different from the problem we focus on. In this paper we propose a method that can find local minima effectively for a general problem 1, while escaping saddle point by stochastic gradient itself depends on some additional assumptions. For example, (Mertikopoulos et al. [2020]) requires the noise of gradient should be uniformly excited. According to our experimental result in Section 5, we can see in some cases stochastic gradient descent cannot escape saddle point effectively or efficiently. Besides, the gradient noise in variance reduced methods is reduced in order to accelerate the convergence. Our experimental results indicate that the gradient noise in variance reduced algorithms is not as good as SGD to serve as the perturbation to avoid saddle point. Therefore, it is necessary to study the second-order stationary point for variance reduced algorithms so as to enable both second-order optimality and fast convergence.

## 3 Method

### 3.1 Algorithm

In this section, we will introduce our PEDESTAL algorithm, which is demonstrated in Algorithm 1. Suppose there are $n$ worker nodes in the decentralized communication network connected by a weight matrix $W$. The initial value of model parameters on all nodes are identical and equal to $x_0$. $x_t^{(i)}$, $v_t^{(i)}$ and $y_t^{(i)}$ are the model parameter, gradient estimator and gradient tracker on the $i$-th worker node in iteration $t$. $z_t^{(i)}$ is the temporary model parameter that is awaiting communication. $\bar{x}_t$, $\bar{v}_t$ and $\bar{y}_t$ are corresponding mean values over all nodes. Counter $esc^{(i)}$ counts the number of iterations in the

---

**Algorithm 1** Perturbed Decentralized STORM Algorithm (PEDESTAL)

---

**Input**: initial value $x_0^{(i)} = x_0$, $v_{-1}^{(i)} = \mathbf{0}$, $y_{-1}^{(i)} = \mathbf{0}$, $esc^{(i)} = -1$.
**Parameter**: $b_0$, $b_1$, $\eta$, $\beta$, $r$, $C_v$, $C_d$, $C_T$.

1: On $i$-th node:
2: **for** $t = 0, 1, \ldots, T - 1$ **do**
3:     **if** $t = 0$ **then**
4:         Compute $v_0^{(i)} = \nabla F_i(x_0, \xi_0^{(i)})$ with $|\xi_0^{(i)}| = b_0$.
5:     **else**
6:         Compute $v_t^{(i)} = \nabla F_i(x_t^{(i)}, \xi_t^{(i)}) + (1 - \beta)(v_{t-1}^{(i)} - \nabla F_i(x_{t-1}^{(i)}, \xi_t^{(i)}))$ with $|\xi_t^{(i)}| = b_1$.
7:     **end if**
8:     Communicate and update the gradient tracker: $y_t^{(i)} = \sum_{j=1}^n w_{ij}(y_{t-1}^{(j)} + v_t^{(j)} - v_{t-1}^{(j)})$.
9:     **if** $esc^{(i)} = -1$ and $\|y_t^{(i)}\| \leq C_v$ **then**
10:        Draw a perturbation $\xi \sim B_0(r)$ and update $z_t^{(i)} = x_t^{(i)} + \xi$.
11:        Save $x_t^{(i)}$ as $\tilde{x}^{(i)}$ and set $esc^{(i)} = 0$.
12:     **else**
13:        Update $z_t^{(i)} = x_t^{(i)} - \eta y_t^{(i)}$.
14:     **end if**
15:     Communicate and update the model parameter: $x_{t+1}^{(i)} = \sum_{j=1}^n w_{ij} z_t^{(j)}$.
16:     **if** $esc^{(i)} \geq 0$ **then**
17:        Reset $esc^{(i)} = -1$ **if** $\|x_{t+1}^{(i)} - \tilde{x}^{(i)}\| > C_d$ **else** update $esc^{(i)} = esc^{(i)} + 1$.
18:     **end if**
19: **end for**
**Return**: $\bar{x}_{t-C_T}$ if there are at least $\frac{n}{10}$ nodes satisfying $esc^{(i)} \geq C_T$.

---

current escaping phase on the $i$-th worker node, which is also the indicator of current phase. When it runs the descent phase on the $i$-th worker node $esc^{(i)}$ is set to $-1$; otherwise $esc^{(i)} \geq 0$.

In the first iteration, the gradient estimator is computed based on a large batch size with $b_0$. Beginning from the second iteration, the gradient estimator $v_t^{(i)}$ is calculated by small mini-batch of samples according to the update rule of STORM, which can be formulated by line 6 in Algorithm 1 where $\beta$ is a hyperparameter of STORM algorithm. Notation $\nabla F_i(x_t^{(i)}, \xi_t^{(i)})$ represents the stochastic gradient obtained from a batch of samples $\xi_t^{(i)}$, which can be written as $\nabla F_i(x_t^{(i)}, \xi_t^{(i)}) = (1/|\xi_t^{(i)}|) \sum_{j \in \xi_t^{(i)}} F_i(x_t^{(i)}, j)$.

After calculating $v_t^{(i)}$, each worker node communicates with its neighbors and update the gradient tracker $y_t^{(i)}$. Inspired by the framework of Perturbed Gradient Descent, our PEDESTAL method also consists of two phases, the descent phase and the escaping phase. If worker node $i$ is in the descent phase and the norm $\|y_t^{(i)}\|$ is smaller than the given threshold $C_v$, then it will draw a perturbation $\xi$ uniformly from $B_0(r)$ and update $z_t^{(i)} = x_t^{(i)} + \xi$. The phase is switched to escaping phase and $esc^{(i)}$ is set to 0. Anchor $\tilde{x}^{(i)} = x_t^{(i)}$ is saved and will be used to discriminate whether the escaping phase is broken. After this iteration counter $esc^{(i)}$ will be added by 1 in each following iteration until the moving distance from the anchor on $i$-th worker node (*i.e.*, $\|x_t^{(i)} - \tilde{x}^{(i)}\|$) is larger than threshold $C_d$ for some $t$, which breaks the escaping phase and turn back to descent phase. If the condition of drawing perturbation is not satisfied, $z_t^{(i)}$ is updated by $z_t^{(i)} = x_t^{(i)} - \eta y_t^{(i)}$ no matter which phase is running currently.

If the $i$-th worker node's counter $esc^{(i)}$ is larger than the threshold $C_T$, it indicates that $\bar{x}_{t-C_T}$ is a candidate second-order stationary point. When at least $\frac{n}{10}$ nodes satisfy the condition $esc^{(i)} \geq C_T$, the algorithm is terminated. Notice that the fraction is set to $\frac{1}{10}$ for convenience in the convergence analysis. Our algorithm also works for other constant fractions. From Algorithm 1 we can see the decision of changing phases on each node only depends on its own status, which is adaptive and independent. Coordinating protocol including broadcast or aggregation is not required.

## 3.2 Discussion

Here we will discuss the insight of the algorithm design and compare the differences between our method and related works. Some novel improvements are the key to the questions in Section 1.

### 3.2.1 Perturbed Gradient Descent or Negative Curvature Descent

Perturbed gradient descent and negative curvature descent are two of the most widely used pure first-order methods to find second-order stationary points. In PEDESTAL algorithm, we adopt the strategy of perturbed gradient descent rather than negative curvature descent because of the following reasons. First, negative curvature descent methods such as Neon (Xu et al. [2018]) and Neon2 (Allen-Zhu and Li [2018]) involves a nested loop to execute the negative curvature subroutine to recognize if a first-order stationary point is a local minimum. However, in decentralized setting, it is possible that the gradient norms on some nodes are smaller than the threshold while others are not. Therefore, some nodes will execute the negative curvature subroutine but its neighbors may not. In this case neighbor nodes need to wait for the nodes running negative curvature subroutines and there will be idle time on neighbor nodes. Besides, the analysis of negative curvature descent methods rely on the precision of the negative curvature direction. It is unknown if the theoretical results are still effective when only a fraction of nodes participate in the computation of negative curvature direction while the others use the gradient. In contrast, perturbed gradient descent only requires a simple operation of drawing perturbation, which is more suitable for decentralized algorithms.

### 3.2.2 Stepsize and Batchsize

In Pullback, a dynamic stepsize $\eta_t = \eta/\|v_t\|$ in the descent phase where $\eta = O(\epsilon)$ and $v_t$ is the gradient estimator. This stepsize is originated from SPIDER (Fang et al. [2018]) which ensures its convergence by bounding the update distance $\|x_{t+1} - x_t\|$. In the escaping phase, Pullback adopts a larger stepsize of $O(1)$ in the escaping phase and a special pullback stepsize in the last iteration, which is the key to improve the gradient complexity. Different from Pullback, in Algorithm 1 we adopt a consistent stepsize such that it keeps invariant even if phase changes and all nodes always use the same stepsize. If there is no perturbation in iteration $t$, we have $\bar{x}_{t+1} = \bar{x}_t - \eta\bar{v}_t$, which is important to the convergence analysis. We discard the strategy in Pullback for two reasons. First, the gradient normalization will probably cause divergence issues in decentralized optimization because in centralized algorithm the gradient direction is $v_t/\|v_t\|$, which is equivalent to $v_c = \sum_{i=1}^n v_t^{(i)}/\|\sum_{i=1}^n v_t^{(i)}\|$. However, in decentralized algorithm the average of $v_t^{(i)}$ is not available on local nodes. If the gradient normalization is done locally, we will get $v_d = \sum_{i=1}^n v_t^{(i)}/\|v_t^{(i)}\|$, which is different to $v_c$ and the error is hard to be estimated. Actually, both D-GET and D-SPIDER-SFO adopt the constant stepsize in SpiderBoost (Wang et al. [2019]) to avoid performing the gradient normalization step. SPIDER needs the gradient normalization because $\|x_{t+1} - x_t\|$ is required to be small in the proof, while SpiderBoost improves the proof to bound $\|x_{t+1} - x_t\|$ by $\eta\|v_t\|$ which is canceled eventually. In our analysis we also adopt the strategy in SpiderBoost. Second, in our algorithm the changing of phase is occurred independently on each node. The phase-wise stepsize and pullback strategy will lead to different stepsizes among all nodes in one iteration, which will also cause potential convergence issues.

In (Chen et al. [2022]), two versions of Pullback are proposed, *i.e.,* Pullback-SPIDER and Pullback-STORM using SPIDER and STORM as the gradient estimator respectively. As introduced previously, one of the advantages of STORM is avoiding large batchsize. Nonetheless, Pullback-STORM adopt a large batchsize of $O(\epsilon^{-1})$ in each iteration, which violates the original intention of STORM. Besides, from Table 1 we can see all algorithms achieving second-order optimality with $\tilde{O}(\epsilon^{-3})$ gradient complexity require a large batchsize of $O(\epsilon^{-1})$. Therefore, we propose a small batch version named PEDESTAL-S as a special case of PEDESTAL that only requires an averaged batchsize of $O(1)$.

### 3.2.3 Conditions of Termination

As a result of applying gradient tracking, we can bound $\frac{1}{n}\sum_{i=1}^n \|y_t^{(i)} - \bar{y}_t\|^2$ by $O(\epsilon^2)$. Even though we have such an estimation, it is still possible that the norm $\|y_t^{(i)}\|$ is as large as $O(\sqrt{n}\epsilon)$ on some nodes when the entire decentralized network has already achieved optimality. Therefore, waiting for all nodes to reach second-order stationary point is not an efficient strategy. This is the reason why we terminate our algorithm when only a fraction of worker nodes satisfy $esc^{(i)} \geq C_T$.

In SSRGD and Pullback, there is an upper bound of iteration numbers in the escaping phase. If the escaping phase is not broken in this number of iterations then the candidate point is regarded as a second-order stationary point. If the escaping phase is broken, then the averaged moving distance is larger than a threshold and the loss function will be reduced by $O(\epsilon^2)$ on average. This strategy guarantees that the algorithm will terminate with a certain gradient complexity. However, in our algorithm worker nodes do not enter escaping phase simultaneously and thus we do not set such an upper bound. In this case the averaged moving distance cannot be lower bounded as $C_T$ has no upper bound. Fortunately, we can complete our analysis by a different novel framework (see the proof outline in the Appendix). An alternative solution is to stop the update on the node that has run certain number of iterations in the escaping phase while the algorithm will continue. But that solution is also challenging since the relation between the first-achieved local optimal solution and the final global optimal solution is unknown and the analysis is non-trivial.

One remaining issue of the current termination strategy is that it involves the global knowledge of how many worker nodes satisfying the termination condition. One solution is to run an additional process to track this global value. The cost of transmitting Boolean values is much less expensive than broadcasting the model. Another solution is to set a maximum iteration in practice. Generally we need to evaluate the model after certain epochs to see if the training process is running smoothly and we can save a checkpoint when we find a better evaluation result. The theoretical analysis ensures that an optimal solution can be visited if the number of iterations is as large as $O(\epsilon^{-3})$.

### 3.2.4 Small Stuck Region

The theoretical guarantee of second-order optimality in SSRGD and Pullback is mainly credit to the lemma of small stuck region, which states that if there are two decoupled sequences $x_t$ and $x_t'$ with identical stochastic samples, $x_s = x_s'$ and $x_{s+1} - x_{s+1}' = r_0 \mathbf{e_1}$ where $\mathbf{e_1}$ is the eigenvector corresponding to the smallest eigenvalue, then it satisfies $\max\{\|x_t - x_s\|, \|x_t' - x_s'\|\} \geq C_d$ for some $s \leq t \leq s + C_T$ with high probability. In SSRGD and Pullback, the averaged moving distance $\frac{1}{t-s} \sum_{\tau=s+1}^{t} \|x_{\tau+1} - x_\tau\|^2$ is used as the criterion to discriminate saddle point because the small stuck region lemma can be applied in this way. However, in decentralized algorithm some nodes enter the escaping phase before the candidate point $\bar{x}_s$ is achieved. Suppose node $i$ enters escaping phase in iteration $s'$, then the averaged moving distance starting from iteration $s$ on node $i$ cannot be well estimated because the condition of not breaking escaping phase on node $i$ only guarantees the bound of averaged moving distance starting from $s'$. Therefore, in our method we use the total moving distance $\|x_t^{(i)} - x_s^{(i)}\|$ as the criterion because we can obtain estimation $\|x_t^{(i)} - x_s^{(i)}\| \leq 2C_d$ given $\|x_t^{(i)} - x_{s'}^{(i)}\| \leq C_d$ and $\|x_s^{(i)} - x_{s'}^{(i)}\| \leq C_d$. And we can further complete our analysis by the small stuck region lemma. Actually we do not require more memory because $\tilde{x}$ is the point to return in SSRGD and Pullback (hence should be saved). In practice, we can also return $\tilde{x}^{(i)}$ for any node $i$ drawing perturbation in iteration $t - C_T$ since $\|x_t^{(i)} - \bar{x}_t\|$ can be well bounded. Besides, we discover that the consensus error $\frac{1}{n} \sum_{i=1}^{n} \|x_t^{(i)} - \bar{x}_t\|^2$ results in an extra term when proving the small stuck region lemma, which becomes another challenge. If the consensus error is not under control, it can drive $x$ away from $x_s$ or push $x$ toward $x_s$, no matter what $\nabla f(x)$ is. In this manner, the stuck region cannot be estimated. In this work, we provide the corresponding proof to estimate this new term exclusively occurred in decentralized setting in our convergence analysis.

## 4    Convergence Analysis

### 4.1    Assumptions

In this section we will provide the main theorem of our convergence analysis. First we will introduce the assumptions used in this paper. All assumptions used in this paper are mild and commonly used in the analysis of related works.

**Assumption 1.** *(Lower Bound) The objective $f$ is lower bounded, i.e., $\inf_x f(x) = f^* > -\infty$.*

**Assumption 2.** *(Bounded Variance) The stochastic gradient of each local loss function is an unbiased estimator and has bounded variance, i.e., for any $i \in \{1, 2, \cdots, n\}$ we have*

$$\mathbb{E}_\xi \nabla F_i(x, \xi) = \nabla f_i(x), \ \mathbb{E}_\xi \|\nabla F_i(x, \xi) - \nabla f_i(x)\|^2 \leq \sigma^2 \tag{2}$$

**Assumption 3.** *(Lipschitz Gradient) For all $\xi$ and $i \in \{1, 2, \cdots, n\}$, $F_i(x, \xi)$ has Lipschitz gradient, i.e., for any $x_1$ and $x_2$ we have $\|\nabla F_i(x_1, \xi) - \nabla F_i(x_2, \xi)\| \leq L\|x_1 - x_2\|$ with a constant $L$.*

**Assumption 4.** *(Lipschitz Hessian) For all $\xi$ and $i \in \{1, 2, \cdots, n\}$, $F_i(x, \xi)$ has Lipschitz hessian, i.e., for any $x_1$ and $x_2$ we have $\|\nabla^2 F_i(x_1, \xi) - \nabla^2 F_i(x_2, \xi)\| \leq \rho\|x_1 - x_2\|$ with a constant $\rho$.*

Assumption 1, Assumption 2 and Assumption 3 are common assumptions used in the analysis of stochastic optimization algorithms. Assumption 4 is the standard assumption to find second-order optimality, which is used in all algorithms that achieves second-order stationary point in Table 1.

**Assumption 5.** *(Spectral Gap) The decentralized network is connected by a doubly-stochastic weight matrix $W \in \mathbb{R}^{n \times n}$ satisfying $W\mathbf{1}_n = W^T\mathbf{1}_n = \mathbf{1}_n$ and $\lambda := \|W - J\| \in [0, 1)$.*

Here $J$ is a $n \times n$ matrix with all elements equal to $\frac{1}{n}$. $W$ is the weight matrix of the decentralized network where $w_{ij} > 0$ if node $i$ and node $j$ are connected, otherwise $w_{ij} = 0$. $\|\cdot\|$ denotes the spectral norm of matrix (*i.e.*, largest singular value). Notice that $\lambda$ is a connectivity measurement of the network graph and it is also the second largest singular value of $W$. We do not assume $W$ to be symmetric and hence the communication network can be both directed graph and undirected graph. The spectral gap assumption is also used commonly in the analysis of decentralized algorithms.

## 4.2 Main Theorems

Let $\epsilon_H = \epsilon^\alpha$. When $\alpha \leq 0.5$, we have the following Theorem 1.

**Theorem 1.** *Assume $\alpha \leq 0.5$ and Assumption 1 to 5 are satisfied. Let $\theta = \min\{\frac{3-5\alpha}{2}, 1\}$. We set $\eta = \Theta(\frac{\epsilon^\theta}{L})$, $\beta = \Theta(\epsilon^{1+\theta})$, $b_0 = \Theta(\epsilon^{-2})$, $b_1 = \Theta(\max\{\epsilon^{2-\theta-5\alpha}, 1\})$, $r = \Theta(\epsilon^{1+\theta})$, $C_v = \Theta(\epsilon)$, $C_T = \tilde{\Theta}(\epsilon^{-\theta-\alpha})$ and $C_d = \tilde{\Theta}(\epsilon^{1-\alpha})$. Then our PEDESTAL algorithm will achieve $O(\epsilon, \epsilon_H)$-second-order stationary point with $\tilde{O}(\epsilon^{-3})$ gradient complexity.*

The specific constants hidden in $\Theta(\cdot)$ will be shown in Appendix B, where the proof outline and the completed proof of Theorem 1 can also be found. From Theorem 1 we can see our PEDESTAL-S with $b_1 = O(1)$ can achieve $O(\epsilon, \epsilon_H)$-second-order stationary point with $\tilde{O}(\epsilon^{-3})$ gradient complexity for $\epsilon_H \geq \epsilon^{0.2}$. In the classic setting, our PEDESTAL achieves second-order stationary point with $\tilde{O}(\epsilon^{-3})$ gradient complexity. When $\alpha > 0.5$, *i.e.*, $\epsilon_H < \sqrt{\epsilon}$, we have the following Theorem 2. Since the parameter settings are different and the $O(1)$ batchsize is only available in Theorem 1, we separate these two theorems. The proof of Theorem 2 can be found in Appendix D.

**Theorem 2.** *When $\epsilon_H < \sqrt{\epsilon}$ (i.e., $\alpha > 0.5$), we set $\eta = \tilde{\Theta}(\epsilon^\theta)$, $\beta = \Theta(\epsilon^{1+\theta})$, $b_0 = \Theta(\epsilon^{-1})$, $b_1 = \tilde{\Theta}(\epsilon^{-\max\{4\alpha-1-\theta, \theta+\alpha\}})$, $r = \Theta(\epsilon^{1+\theta})$, $C_v = \Theta(\epsilon)$, $C_T = \tilde{\Theta}(\epsilon^{-\theta-\alpha})$ and $C_d = \tilde{\Theta}(\epsilon^\alpha)$ where $\theta = \min\{\frac{3\alpha-1}{2}, 3\alpha - 2\}$. Under Assumption 1 to 5, our PEDESTAL algorithm will achieve $O(\epsilon, \epsilon_H)$-second-order stationary point with $\tilde{O}(\epsilon\epsilon_H^{-8} + \epsilon^4\epsilon_H^{-11})$ gradient complexity.*

# 5 Experiments

In this section we will demonstrate our experimental results to validate the performance of our method. We conduct two tasks in our experiment, a matrix sensing task on synthetic dataset and a matrix factorization task on real-world dataset. Both of these two tasks are non-spurious local minimum problems (Ge et al. [2017, 2016]), which means all local minima are global minima. Thus, we conclude an algorithm is stuck at saddle point if the loss function value does not achieve the global minimum. The source code is available in `https://github.com/WH-XIAN/PEDESTAL`.

## 5.1 Matrix Sensing

We follow the experimental setup of (Chen et al. [2022]) to solve a decentralized matrix sensing problem. The goal of this task is to recover a low-rank $d \times d$ symmetric matrix $M^* = U^*(U^*)^T$ where $U^* \in \mathbb{R}^{d \times r}$ for some small $r$. We set the number of worker nodes to $n = 20$. We generate a synthetic dataset with $N$ sensing matrices $\{A_i\}_{i=1}^N$ and $N$ corresponding observations $b_i = \langle A_i, M^* \rangle$. Here the inner product $\langle X, Y \rangle$ of two matrices $X$ and $Y$ is defined by the trace $tr(X^T Y)$. The decentralized optimization problem can be formulated by

$$\min_{U \in \mathbb{R}^{d \times r}} \sum_{i=1}^n L_i(U), \text{ where } L_i(U) = \frac{1}{2}\sum_{j=1}^{N_i}(\langle A_{ij}, UU^T \rangle - b_{ij})^2, \tag{3}$$

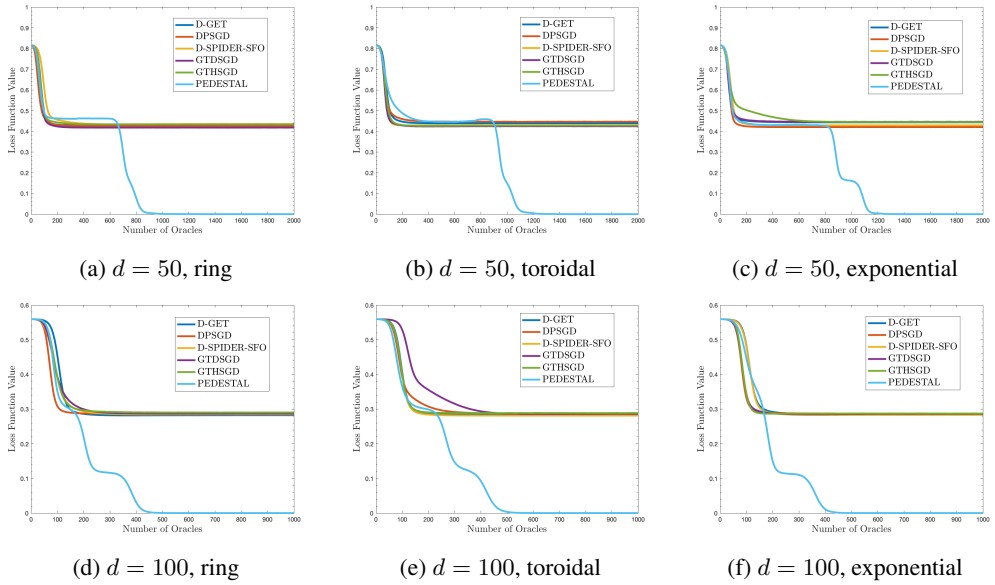

(a) $d = 50$, ring        (b) $d = 50$, toroidal        (c) $d = 50$, exponential

(d) $d = 100$, ring        (e) $d = 100$, toroidal        (f) $d = 100$, exponential

Figure 1: Experimental results of the decentralized matrix sensing task on different network topology for $d = 50$ and $d = 100$. Data is assigned to worker nodes by random distribution. The y-axis is the loss function value and the x-axis is the number of gradient oracles divided by the number of data $N$.

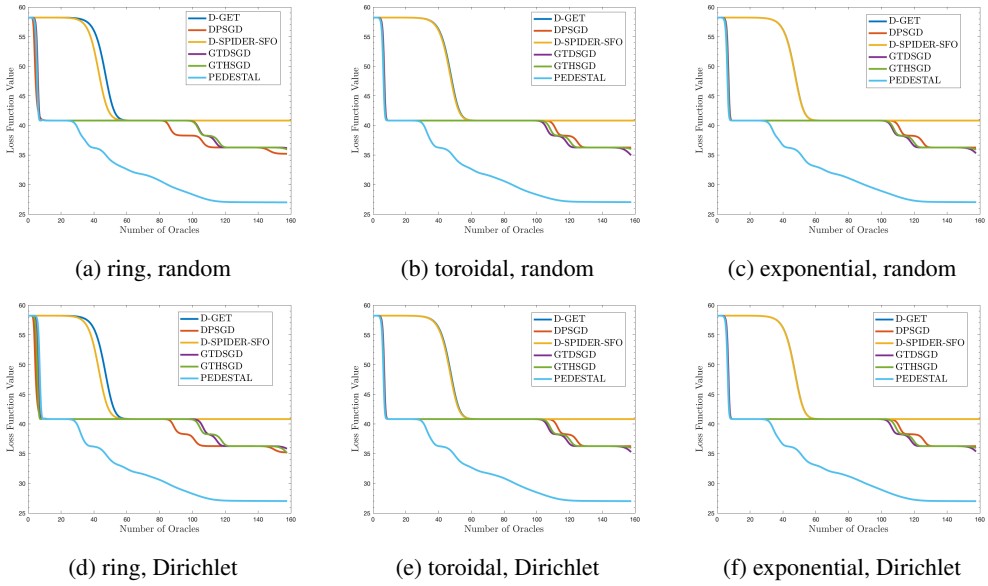

(a) ring, random        (b) toroidal, random        (c) exponential, random

(d) ring, Dirichlet        (e) toroidal, Dirichlet        (f) exponential, Dirichlet

Figure 2: Experimental results of the decentralized matrix factorization task on different network topology on MovieLens-100k. The y-axis is the loss function value and the x-axis is the number of gradient oracles divided by the size of matrix $N \times l$.

where $N_i$ is the amount of data assigned to worker node $i$.

The number of rows of matrix $U$ is set to $d = 50$ and $d = 100$ respectively and the number of columns is set to $r = 3$. The ground truth low-rank matrix $M^*$ equals $U^*(U^*)^T$ where each entry of $U^*$ is generated by Gaussian distribution $\mathcal{N}(0, 1/d)$ independently. We randomly generate $N = 20 \times n \times d$ samples of sensing matrices $\{A_i\}_{i=1}^N$, $A_i \in \mathbb{R}^{d \times d}$ from standard Gaussian distribution and calculate the corresponding labels $b_i = \langle A_i, M^* \rangle$. We consider two different types of data distribution, random distribution and Dirichlet distribution $Dir_{20}(0.3)$ to assign data to each worker node. We conduct experiments on three different types of network topology, *i.e.*, ring topology, toroidal

topology (2-dimensional ring) and undirected exponential graph. The initial value of $U$ is set to $[u_0, \mathbf{0}, \mathbf{0}]$ where $u_0$ is yield from Gaussian distribution and multiplied by a scalar such that it satisfies $\|u_0\| \leq \max eig(M^*)$. We compare our PEDESTAL algorithm to decentralized baselines including D-PSGD, GTDSGD, D-GET, D-SPIDER-SFO and GTHSGD. In this experiment, the learning rate is chosen from $\{0.01, 0.001, 0.0001\}$. The batchsize is set to 10. For PEDESTAL and GTHSGD, parameter $\beta$ is set to 0.01. For D-GET and D-SPIDER-SFO, the period $q$ is 100. For PEDESTAL, threshold $C_v$ is set to 0.0001. Perturbation radius $r$ is set to 0.001. The threshold of moving distance $C_d$ is set to 0.01. The experimental results are shown in Figure 1. Due to the space limit, we only show the result of random data distribution in the main manuscript and leave the result of Dirichlet distribution to Appendix A.

From the experimental result we can see all baselines are stuck at the saddle point and cannot escape it effectively. In contrast, our PEDESTAL reaches and escapes saddle points and finally find the local minimum. We also calculate the smallest eigenvalue of hessian matrix for each algorithm at the converged optimal point, which is left to the Supplementary Material because of space limit. According to the eigenvalue result, we can see the smallest eigenvalue is much closer to 0 than all baselines. Therefore, our experiment verifies that our PEDESTAL achieves the best performance to escape saddle point and find local minimum.

## 5.2 Matrix Factorization

The second task in our experiment is matrix factorization, which aims to approximate a given matrix $M \in \mathbb{R}^{N \times l}$ by a low-rank matrix that can be decomposed to the product of two matrices $U \in \mathbb{R}^{N \times r}$ and $V \in \mathbb{R}^{l \times r}$ for some small $r$. The optimization problem can be formulated by

$$\min_{U \in \mathbb{R}^{N \times r}, V \in \mathbb{R}^{l \times r}} \|M - UV^T\|_F^2 := \sum_{i=1}^{N} \sum_{j=1}^{l} (M_{ij} - (UV^T)_{ij})^2 \tag{4}$$

where $\| \cdot \|_F$ denotes the Frobenius norm and subscript $ij$ refers to the element at $i$-th row and $j$-th column. In our experiment we solve this problem on the MovieLens-100k dataset (Harper and Konstan [2015]). MovieLens-100k contains 100,000 ratings of 1682 movies provided by 943 users. Each rating is in the interval $[0, 5]$ and scaled to $[0, 1]$ in the experiment. This task can be regarded as an association task to predict users' potential ratings for unseen movies. In our experiment we set the number of worker node to $n = 50$. Each node is assigned the data from different group of users. Similar to the matrix sensing task, here we also use random distribution and Dirichlet distribution respectively to distribute users to worker nodes. And we also use ring topology, toroidal topology and undirected exponential graph as the communication network. The baselines are also D-PSGD, GTDSGD, D-GET, D-SPIDER-SFO and GTHSGD. In this experiment, the number of worker nodes is 50 and the rank of the matrix $M$ is set to 25. The learning rate is chosen from $\{0.01, 0.001, 0.0001\}$. The batchsize is set to 100. For PEDESTAL and GTHSGD, parameter $\beta$ is set to 0.1. For D-GET and D-SPIDER-SFO, the period $q$ is 100. For PEDESTAL, threshold $C_v$ is set to 0.002. Perturbation radius $r$ is set to 0.01. The threshold of moving distance $C_d$ is set to 0.5. The experimental results are shown in Figure 2.

From the experimental results we can see our PEDESTAL achieves the best performance to escape saddle point and find second-order stationary point. All baselines cannot escape saddle point effectively or efficiently. Particularly, variance reduced methods D-GET and D-SPIDER-SFO shows worse performance than SGD based algorithms D-PSGD and GTDSGD, which indicates that although reducing gradient noise can accelerate convergence, it also weakens the ability to escape saddle point. Therefore, our contribution is important since we make the fast convergence of variance reduction compatible with the capability to avoid saddle point.

## 6 Conclusion

In this paper we propose a novel algorithm PEDESTAL to find local minima in nonconvex decentralized optimization. PEDESTAL is the first decentralized stochastic algorithm to achieve second-order optimality with non-asymptotic analysis. We improve the drawbacks in previous deterministic counterpart to make phase changed independently on each node and avoid consensus protocols of broadcast or aggregation. We prove that PEDESTAL can achieve $O(\epsilon, \sqrt{\epsilon})$-second-order stationary point with the gradient complexity of $\tilde{O}(\epsilon^{-3})$, which matches state-of-the-art results of centralized counterpart or decentralized method to find first-order stationary point. We also conduct the matrix sensing and matrix factorization tasks in our experiments to validate the performance of PEDESTAL.

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

# A    Additional Experimental Results

The experimental results of Dirichlet distribution of the matrix sensing task is shown in Figure 3. The smallest eigenvalue at the converged point for each algorithm is shown in Table 2 and Table 3.

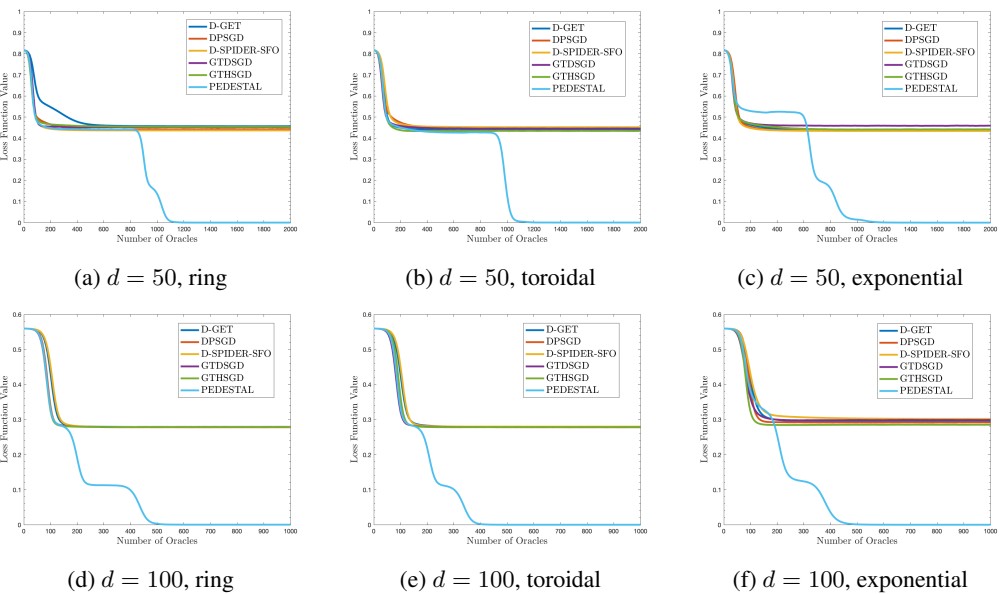

(a) $d = 50$, ring          (b) $d = 50$, toroidal          (c) $d = 50$, exponential

(d) $d = 100$, ring          (e) $d = 100$, toroidal          (f) $d = 100$, exponential

Figure 3: Experimental results of the decentralized matrix sensing task on different network topology for $d = 50$ and $d = 100$. Data is assigned to worker nodes by Dirichlet distribution. The y-axis is the loss function value and the x-axis is the number of gradient oracles divided by the number of data $N$.

|  | D-PSGD | GTDSGD | D-GET | D-SPIDER-SFO | GTHSGD | PEDESTAL |
|---|---|---|---|---|---|---|
| $d = 50$, ring | -0.0332 | -0.0327 | -0.0333 | -0.0328 | -0.0329 | $\mathbf{-1.78e^{-5}}$ |
| $d = 50$, toroidal | -0.0331 | -0.0334 | -0.0334 | -0.0327 | -0.0329 | $\mathbf{-4.18e^{-5}}$ |
| $d = 50$, exponential | -0.0323 | -0.0330 | -0.0331 | -0.0332 | -0.0333 | $\mathbf{-1.09e^{-6}}$ |
| $d = 100$, ring | -0.0184 | -0.0184 | -0.0184 | -0.0184 | -0.0185 | $\mathbf{-2.07e^{-6}}$ |
| $d = 100$, toroidal | -0.0185 | -0.0186 | -0.0185 | -0.0184 | -0.0184 | $\mathbf{-2.25e^{-7}}$ |
| $d = 100$, exponential | -0.0184 | -0.0184 | -0.0186 | -0.0184 | -0.0184 | $\mathbf{-3.07e^{-5}}$ |

Table 2: Smallest eigenvalue of hessian matrix at the converged point (random data distribution).

|  | D-PSGD | GTDSGD | D-GET | D-SPIDER-SFO | GTHSGD | PEDESTAL |
|---|---|---|---|---|---|---|
| $d = 50$, ring | -0.0332 | -0.0337 | -0.0332 | -0.0325 | -0.0330 | $\mathbf{-3.60e^{-6}}$ |
| $d = 50$, toroidal | -0.0334 | -0.0324 | -0.0329 | -0.0325 | -0.0327 | $\mathbf{-2.29e^{-5}}$ |
| $d = 50$, exponential | -0.0334 | -0.0326 | -0.0333 | -0.0330 | -0.0328 | $\mathbf{-3.97e^{-5}}$ |
| $d = 100$, ring | -0.0184 | -0.0184 | -0.0184 | -0.0185 | -0.0183 | $\mathbf{-4.48e^{-5}}$ |
| $d = 100$, toroidal | -0.0184 | -0.0184 | -0.0184 | -0.0184 | -0.0185 | $\mathbf{-1.24e^{-5}}$ |
| $d = 100$, exponential | -0.0186 | -0.0185 | -0.0186 | -0.0183 | -0.0185 | $\mathbf{-3.63e^{-6}}$ |

Table 3: Smallest eigenvalue of hessian matrix at the converged point (Dirichlet data distribution).

# B    Proof of Theorem 1

## B.1    Notation

We define matrix $X_t = [x_t^{(1)}, \cdots, x_t^{(n)}] \in \mathbb{R}^{d \times n}$ where $x_t^{(i)}$ is the model parameter on $i$-th worker node with dimension $d$ and $n$ is the number of worker nodes. Similarly we have $Y_t = [y_t^{(1)}, \cdots, y_t^{(n)}]$,

$Z_t = [z_t^{(1)}, \cdots, z_t^{(n)}]$ and $V_t = [v_t^{(1)}, \cdots, v_t^{(n)}]$. Let $\omega_t = \|\bar{x}_{t+1} - \bar{x}_t\|^2$ and $\Omega_t = Z_t - X_t$. Define $p_t = n_t/n$ where $n_t$ is the number of worker nodes drawing perturbation in iteration $t$.

## B.2 Outline

In this section we will provide the proof outline of Theorem 1. First, we prove some basic lemmas to estimate gradient noise and consensus error, which will be used frequently in later proof. The gradient noise is estimated by Lemma 1, the proof of which can be found in Section C.1. The consensus error is estimated by Lemma 2, the proof of which can be found in Section C.2.

**Lemma 1.** *(Gradient Noise) Under Assumption 2 and Assumption 3 we have*

$$(a) \quad \frac{1}{nT}\sum_{t=0}^{T-1}\sum_{i=1}^{n}\|v_t^{(i)} - \nabla f_i(x_t^{(i)})\|^2 \leq \frac{16\log(4/\delta)\beta\sigma^2}{b_1} + \frac{384\log(4/\delta)L^2}{nb_1\beta T}\sum_{t=0}^{T-1}\|X_t - \bar{X}_t\|_F^2$$

$$+ \frac{192\log(4/\delta)L^2}{b_1\beta T}\sum_{t=0}^{T-1}\omega_t + \frac{2\log(4/\delta)\sigma^2}{\beta b_0 T}$$

$$(b) \quad \frac{1}{T}\sum_{t=1}^{T}\|\bar{v}_t - \frac{1}{n}\sum_{i=1}^{n}\nabla f_i(x_t^{(i)})\|^2 \leq \frac{16\log(4/\delta)\beta\sigma^2}{nb_1} + \frac{384\log(4/\delta)L^2}{n^2 b_1\beta T}\sum_{t=1}^{T}\|X_t - \bar{X}_t\|_F^2$$

$$+ \frac{192\log(4/\delta)L^2}{nb_1\beta T}\sum_{t=0}^{T-1}\omega_t + \frac{2\log(4/\delta)\sigma^2}{n\beta b_0 T}$$

**Lemma 2.** *(Consensus Error) Let $\eta \leq \frac{(1-\lambda)^2\epsilon^\theta}{600\log(4/\delta)\lambda^2 L}$, $\beta = C_1^{-1}\epsilon^{1+\theta}$ and $b_1 \geq C_1\epsilon^{-1+\theta}$ where $C_1 \geq 1$ is a constant. Under Assumption 2, 3 and 5 we have*

$$(a) \quad \frac{1}{T}\sum_{t=1}^{T}\|X_t - \bar{X}_t\|_F^2 \leq \frac{160000n\log(4/\delta)L^2\eta^2\lambda^4}{(1-\lambda)^4\min\{b_1\beta,1\}T}\sum_{t=0}^{T-1}\omega_t + \frac{12288n\log(4/\delta)\beta\eta^2\lambda^4\sigma^2}{(1-\lambda)^4 b_1}$$

$$+ \frac{2000n\log(4/\delta)\eta^2\lambda^4\sigma^2}{(1-\lambda)^4\beta b_0 T} + \frac{128\lambda^4\eta^2}{(1-\lambda)^3 T}\sum_{i=1}^{n}\|\nabla f_i(x_0)\|^2 + \sum_{t=0}^{T-1}\frac{64n\lambda^2 p_t(\eta^2 C_v^2 + r^2)}{(1-\lambda)^2 T}$$

$$(b) \quad \frac{1}{T}\sum_{t=0}^{T-1}\|Y_t - \bar{Y}_t\|_F^2 \leq \frac{4644\log(4/\delta)nL^2\lambda^2}{(1-\lambda)\min\{b_1\beta,1\}T}\sum_{t=0}^{T-1}\omega_t + \frac{384\log(4/\delta)n\lambda^2\beta\sigma^2}{(1-\lambda)b_1}$$

$$+ \frac{50\log(4/\delta)n\lambda^2\sigma^2}{(1-\lambda)\beta b_0 T} + \frac{8\lambda^2}{(1-\lambda)T}\sum_{i=1}^{n}\|\nabla f_i(x_0)\|^2 + \sum_{t=0}^{T-1}\frac{150000\log(4/\delta)nL^2\lambda^4 p_t(\eta^2 C_v^2 + r^2)}{(1-\lambda)^3\min\{b_1\beta,1\}T}$$

Next we will prove that PEDESTAL will terminate in certain number of iterations. Under Assumption 2, 3 and 5, we can prove the following Lemma 3. The proof is demonstrated in Section C.3.

**Lemma 3.** *(Descent) Let $\eta \leq \frac{(1-\lambda)^2\epsilon^\theta}{600\log(4/\delta)\lambda^2 L}$, $\beta = C_1^{-1}\epsilon^{1+\theta}$, $b_1 \geq C_1\epsilon^{-1+\theta}$ and $b_0 = C_1\epsilon^{-1}$ where $C_1 \geq 1$ is a constant. Under Assumption 2, 3 and 5 we have*

$$f(\bar{x}_T) \leq f(x_0) + \frac{\sigma^2}{L} + \frac{1}{nL}\sum_{i=1}^{n}\|\nabla f_i(x_0)\|^2 - \sum_{t=0}^{T-1}\mathcal{D}_t$$

*where*

$$\mathcal{D}_t = \frac{1}{16\eta}\omega_t + \frac{(1-\lambda)^2}{256n\eta}\sum_{i=1}^{n}\|x_{t+1}^{(i)} - x_t^{(i)}\|^2 + \frac{\eta}{2n}\sum_{i=1}^{n}\|y_t^{(i)}\|^2 - \frac{200\eta\epsilon^2\sigma^2}{(1-\lambda)^2 C_1^2} - \frac{7p_t(\eta^2 C_v^2 + r^2)}{4\eta}$$

Here we call $\mathcal{D}_t$ the descent of iteration $t$. We categorize all iterations into three types:

$$\text{type-A: } p_t \geq \frac{1}{5}, \quad \text{type-B: } p_t < \frac{1}{5} \text{ and } \frac{1}{n}\sum_{i=1}^{n}\|y_t^{(i)}\|^2 \geq \frac{4C_v^2}{5}, \quad \text{type-C: otherwise}$$

When at least $\frac{n}{5}$ nodes drawing perturbation in iteration $t$, then it is type-A. There are two cases where $p_t$ is small: most nodes in the descent phase or most nodes in the escaping phase. An iteration is type-B if $p_t < \frac{1}{5}$ and $\frac{1}{n}\sum_{i=1}^{n}\|y_t^{(i)}\|^2 \geq \frac{4C_v^2}{5}$, which represents the case where most nodes are in the descent phase. And type-C iteration represents the case where most nodes are in the escaping phase. Next we will estimate type-A and type-C iteration with the following Lemma 4.

**Lemma 4.** *Let* $\eta \leq \frac{(1-\lambda)^2\epsilon^\theta}{600\log(4/\delta)\lambda^2 L}$, $\beta = C_1^{-1}\epsilon^{1+\theta}$, $b_1 \geq C_1\epsilon^{-1+\theta}$, $b_0 = C_1\epsilon^{-1}$, $C_d = C_2\eta C_T\epsilon$, $C_v = \frac{(1-\lambda)C_2\epsilon}{200}$ *and* $r \leq \eta C_v/4$ *where* $C_1 = \frac{20000\sigma}{(1-\lambda)^2 C_2}$ *and* $C_2$ *is a constant. Under Assumption 2, 3 and 5, we can find disjoint intervals* $\mathcal{I} = \mathcal{I}_1 \cup \cdots \cup \mathcal{I}_k$ *such that the indexes of all type-A and type-C iterations except the last* $C_T$ *iterations are contained in* $\mathcal{I}$ *and the descent over* $\mathcal{I}$ *can be estimated by*

$$\sum_{t\in\mathcal{I}}\mathcal{D}_t \geq |\mathcal{I}| \cdot \frac{(1-\lambda)^2 C_2^2 \eta\epsilon^2}{10000}$$

*where* $|\cdot|$ *denotes the total number of the set.*

Besides, for all type-B iteration $t$, we have the following estimation

**Lemma 5.** *Let parameter and assumption settings be the same as Lemma 4, then for all type-B iteration* $t$ *we have*

$$\mathcal{D}_t \geq \frac{(1-\lambda)^2 C_2^2 \eta\epsilon^2}{8000000}$$

With Lemma 4, Lemma 5 and Assumption 1, we can conclude that PEDESTAL will terminate in $\tilde{O}(\epsilon^{-2-\theta}) + C_T$ iterations. As the last two negative terms in $\mathcal{D}_t$ are canceled by $\frac{1}{n}\sum_{i=1}^{n}\|x_{t+1}^{(i)} - x_t^{(i)}\|^2$ and $\frac{1}{n}\sum_{i=1}^{n}\|y_t^{(i)}\|^2$ respectively in Lemma 4 and Lemma 5, we have $\frac{1}{\eta}\sum_{t=0}^{T-1}\omega_t \leq O(1)$. Hence by Lemma 2 we know the consensus error $\frac{1}{n}\|X_t - \bar{X}_t\|_F^2$ can be bounded by $O(\epsilon^{1+\theta})$ on average. Besides, from the parameter setting we can see $C_v$ is $\Theta(\epsilon)$, which ensures the first-order optimality of the decentralized algorithm.

Finally, we will prove PEDESTAL is able to achieve second-order stationary point. First, we will give the small stuck region lemma in decentralized setting. Recall that $\epsilon_H = \epsilon^\alpha$ is the tolerance of second-order stationary point. The proof is in Section C.6.

**Lemma 6.** *(Small Stuck Region) Suppose* $n_s$ *worker nodes draw perturbation in iteration* $s$ *and* $-\gamma = \min eig(\nabla^2 f(\bar{x}_s)) \leq -\epsilon_H$. *Let* $\eta \leq \frac{(1-\lambda)^2\epsilon^\theta}{1000\sqrt{n}\log(C_T)\log(4/\delta)\lambda^2 L}$, $\beta = C_1^{-1}\epsilon^{1+\theta}$, $b_1 \geq 1000C_1\epsilon^{2-\theta-5\alpha}$, $C_d = C_2\eta C_T\epsilon^\mu$ *and* $C_T = \log(12nC_d/r_0)/(\eta\gamma)$ *where* $C_1 = \frac{20000}{(1-\lambda)^2 C_2}$, $C_2 \leq \frac{1-\lambda}{2000\log(4/\delta)\rho\log(C_d)}$ *and* $\mu = \max\{1, 2\alpha\}$. *Let* $X_t$ *and* $X_t'$ *be two coupled decentralized sequences by running PEDESTAL from* $X_s$ *with* $X_s = X_s'$, $x_{s+1}^{(i)} = x_{s+1}^{(i)'}$ *if node* $i$ *does not draw perturbation in iteration* $s$ *and* $x_{s+1}^{(i)} = x_{s+1}^{(i)'} + r_0\mathbf{e_1}$ *otherwise. Here* $\mathbf{e_1}$ *is the eigenvector with respect to the smallest eigenvalue* $\gamma$. *Define* $d_i = \max_{s\leq t\leq s+C_T}\{\|x_t^{(i)} - x_s^{(i)}\|, \|x_t^{(i)'} - x_s^{(i)}\|\}$. *Then there are at least* $\frac{9n}{10}$ *nodes such that* $d_i \geq 2C_d$.

In decentralized small stuck region lemma, the consensus error will lead to a new term (see Eq. (34)) and make the proof more complicated. In our proof, we use the condition of $\epsilon_H \geq \epsilon$, *i.e.*, $\alpha \leq 1$. For smaller $\epsilon_H$ the batchsize $b_1$ is required to set larger. With Lemma 6, we can prove that when PEDESTAL is terminated, it finds a local minimum with high probability.

**Lemma 7.** *Let* $r_0 = \delta r/\sqrt{d}$ *where* $d$ *is the dimension of model parameter. Other parameters are the same as Lemma 6. Suppose PEDESTAL is terminated in iteration* $s + C_T$. *Then* $\bar{x}_s$ *is a second-order stationary point with probability at least* $1 - \delta$.

Lemma 7 provides the guarantee of second-order optimality of PEDESTAL. When $\epsilon_H \geq \sqrt{\epsilon}$, *i.e.*, $\alpha \leq 0.5$ (including the classic setting $\epsilon_H = \sqrt{\epsilon}$), the parameter settings of all lemmas are consistent and the main theorem is proven. The total gradient complexity is

$$\tilde{O}(\epsilon^{-2-\theta} \cdot \epsilon^{-1+\theta}) = \tilde{O}(\epsilon^{-3})$$

When $\alpha = 0.5$, we have $\theta = 0.25$ and $b_1 = \Theta(\epsilon^{-0.75})$. When $\alpha \le 0.2$, we can set $\theta = 1$ and $b_1 = O(1)$, which is result of PEDESTAL-S. In Section D we will provide the analysis of the case $\alpha > 0.5$ with a different parameter setting of $\theta$ and $b_1$. We can achieve the gradient complexity of

$$\tilde{O}(\epsilon^{-3} + \epsilon\epsilon_H^{-8} + \epsilon^4\epsilon_H^{-11}) \tag{5}$$

over all cases of $\epsilon_H$.

## C Proof of Lemmas

### C.1 Proof of Lemma 1

*Proof.* According to the definition of $v_t^{(i)}$, we have

$$\frac{v_{t+1}^{(i)} - \nabla f_i(x_{t+1}^{(i)})}{(1-\beta)^{t+1}} - \frac{v_t^{(i)} - \nabla f_i(x_t^{(i)})}{(1-\beta)^t} = \frac{\beta(\nabla F_i(x_{t+1}^{(i)}, \xi_{t+1}^{(i)}) - \nabla f_i(x_{t+1}^{(i)}))}{(1-\beta)^{t+1}}$$
$$+ \frac{(\nabla F_i(x_{t+1}^{(i)}, \xi_{t+1}^{(i)}) - \nabla f_i(x_{t+1}^{(i)})) - (\nabla F_i(x_t^{(i)}, \xi_{t+1}^{(i)}) - \nabla f_i(x_t^{(i)}))}{(1-\beta)^t} \tag{6}$$

where $|\xi_{t+1}^{(i)}| = b_1$. The expectation of the right side of Eq. (6) over $\xi_{t+1}^{(i)}$ is 0. Using Cauchy-Schwartz inequality, Assumption 2 and Assumption 3 we have

$$\|\frac{\beta(\nabla F_i(x_{t+1}^{(i)}, j) - \nabla f_i(x_{t+1}^{(i)}))}{(1-\beta)^{t+1}} + \frac{(\nabla F_i(x_{t+1}^{(i)}, j) - \nabla f_i(x_{t+1}^{(i)})) - (\nabla F_i(x_t^{(i)}, j) - \nabla f_i(x_t^{(i)}))}{(1-\beta)^t}\|^2$$
$$\le \frac{2\beta^2\sigma^2}{(1-\beta)^{2t+2}} + \frac{8L^2\|x_{t+1}^{(i)} - x_t^{(i)}\|^2}{(1-\beta)^{2t}} \tag{7}$$

for each $j \in \xi_{t+1}^{(i)}$. Thus, applying Azuma-Hoeffding inequality to Eq. (6) we can obtain

$$\|v_t^{(i)} - \nabla f_i(x_t^{(i)}) - (1-\beta)^t(v_0^{(i)} - \nabla f_i(x_0))\|^2$$
$$\le \frac{4\log(4/\delta)}{b_1}(2\beta^2\sigma^2 + 8L^2\sum_{s=0}^{t-1}(1-\beta)^{2(t-s)}\|x_{s+1}^{(i)} - x_s^{(i)}\|^2) \tag{8}$$

with probability $1 - \delta$. Here we use the fact that $\sum_{s=0}^{+\infty}(1-\beta)^s = \frac{1}{\beta}$. Using Cauchy-Schwartz inequality to Eq. (8) we have

$$\|v_t^{(i)} - \nabla f_i(x_t^{(i)})\|^2 \le \frac{16\log(4/\delta)}{b_1}(\beta\sigma^2 + 4L^2\sum_{s=0}^{t-1}(1-\beta)^{2(t-s)}\|x_{s+1}^{(i)} - x_s^{(i)}\|^2)$$
$$+ 2(1-\beta)^{2t}\|v_0^{(i)} - \nabla f_i(x_0)\|^2 \tag{9}$$

Sum Eq. (9), we obtain

$$\frac{1}{\log(4/\delta)nT}\sum_{t=0}^{T-1}\sum_{i=1}^{n}\|v_t^{(i)} - \nabla f_i(x_t^{(i)})\|^2$$
$$\le \frac{16\beta\sigma^2}{b_1} + \frac{64L^2}{nb_1\beta T}\sum_{t=0}^{T-2}\|X_{t+1} - X_t\|_F^2 + \frac{2\sigma^2}{\beta b_0 T}$$
$$\le \frac{16\beta\sigma^2}{b_1} + \frac{384L^2}{nb_1\beta T}\sum_{t=0}^{T-1}\|X_t - \bar{X}_t\|_F^2 + \frac{192L^2}{b_1\beta T}\sum_{t=0}^{T-1}\omega_t + \frac{2\sigma^2}{\beta b_0 T} \tag{10}$$

which finishes the proof of (a) in Lemma 1. In the first inequality of Eq. (10) we apply Azuma-Hoeffding inequality to $v_0^{(i)} - \nabla f_i(x_0)$. In the second inequality we apply Cauchy-Schwartz inequality and use the fact $x_{t+1}^{(i)} - x_t^{(i)} = (x_{t+1}^{(i)} - \bar{x}_{t+1}) - (x_t^{(i)} - \bar{x}_t) + (\bar{x}_{t+1} - \bar{x}_t)$. Mimic above steps and we can achieve the inequality (b) in Lemma 1. The term $n$ in the denominator is derived by the fact that $\xi_t^{(i)}$'s on different nodes are independent. $\qquad\square$

## C.2 Proof of Lemma 2

*Proof.* As $Y_t = W(Y_{t-1} + V_t - V_{t-1})$, we have

$$
\|Y_t - \bar{Y}_t\|_F^2
$$
$$
= \|(W - J)(Y_{t-1} - \bar{Y}_{t-1}) + (W - J)(V_t - V_{t-1})\|_F^2
$$
$$
\leq \lambda^2 \|Y_{t-1} - \bar{Y}_{t-1}\|_F^2 + 2\langle (W - J)Y_t, (W - J)(V_t - V_{t-1})\rangle + \lambda^2 \|V_t - V_{t-1}\|_F^2
$$
$$
\leq \frac{1 + \lambda^2}{2}\|Y_{t-1} - \bar{Y}_{t-1}\|_F^2 + \frac{\lambda^2 + \lambda^4}{1 - \lambda^2}\|V_t - V_{t-1}\|_F^2
$$
$$
\leq \frac{1 + \lambda^2}{2}\|Y_{t-1} - \bar{Y}_{t-1}\|_F^2 + \frac{3\lambda^2(1 + \lambda^2)}{1 - \lambda^2}\sum_{i=1}^{n}(\|v_t^{(i)} - \nabla f_i(x_t^{(i)})\|^2 + \|v_{t-1}^{(i)} - \nabla f_i(x_{t-1}^{(i)})\|^2)
$$
$$
+ \frac{9L^2\lambda^2(1 + \lambda^2)}{1 - \lambda^2}(\|X_t - \bar{X}_t\|_F^2 + \|X_{t-1} - \bar{X}_{t-1}\|_F^2 + n\omega_{t-1}) \tag{11}
$$

where the first inequality is derived by Assumption 5, the second inequality is derived by Young's inequality and the last inequality is derived by Cauchy-Schwartz inequality and Assumption 3. When $t = 0$, by Azuma-Hoeffding inequality we can get

$$
\|Y_0 - \bar{Y}_0\|_F^2 \leq 2\lambda^2 \sum_{i=1}^{n} \|\nabla f_i(x_0)\|^2 + \frac{8\log(4/\delta)n\lambda^2\sigma^2}{b_0} \tag{12}
$$

with probability $1 - \delta$. As $X_{t+1} = W(X_t + \Omega_t)$, by Assumption 5 and Young's inequality we have

$$
\|X_{t+1} - \bar{X}_{t+1}\|_F^2
$$
$$
\leq \frac{1 + \lambda^2}{2}\|X_t - \bar{X}_t\|_F^2 + \frac{2\lambda^2}{1 - \lambda^2}\|\Omega_t - \bar{\Omega}_t\|_F^2
$$
$$
\leq \frac{1 + \lambda^2}{2}\|X_t - \bar{X}_t\|_F^2 + \frac{4\eta^2\lambda^2}{1 - \lambda^2}\|Y_t - \bar{Y}_t\|_F^2 + \frac{4\lambda^2}{1 - \lambda^2}\|\Omega_t - \bar{\Omega}_t - \eta(Y_t - \bar{Y}_t)\|_F^2
$$
$$
\leq \frac{1 + \lambda^2}{2}\|X_t - \bar{X}_t\|_F^2 + \frac{4\eta^2\lambda^2}{1 - \lambda^2}\|Y_t - \bar{Y}_t\|_F^2 + \frac{8n\lambda^2 p_t(\eta^2 C_v^2 + r^2)}{1 - \lambda^2} \tag{13}
$$

where the second inequality is obtained by Cauchy-Schwartz inequality and the last inequality is because when node $i$ draws perturbation it must satisfy $\|y_t^{(i)}\| \leq C_v$. Note that $X_0 = \bar{X}_0$. Sum Eq. (13), we have

$$
\sum_{t=1}^{T} \|X_t - \bar{X}_t\|_F^2
$$
$$
\leq \frac{8\eta^2\lambda^2}{(1 - \lambda^2)^2}\sum_{t=0}^{T-1} \|Y_t - \bar{Y}_t\|_F^2 + \frac{16n\lambda^2(\eta^2 C_v^2 + r^2)T}{(1 - \lambda^2)^2}
$$
$$
\leq \frac{288L^2\eta^2\lambda^4(1 + \lambda^2)}{(1 - \lambda^2)^4}\sum_{t=0}^{T-1} \|X_t - \bar{X}_t\|_F^2 + \frac{96\eta^2\lambda^4(1 + \lambda^2)}{(1 - \lambda^2)^4}\sum_{t=0}^{T-1}\sum_{i=1}^{n} \|v_t^{(i)} - \nabla f_i(x_t^{(i)})\|^2
$$
$$
+ \frac{144nL^2\eta^2\lambda^4(1 + \lambda^2)}{(1 - \lambda^2)^4}\sum_{t=0}^{T-1} \omega_t + \frac{16\lambda^2\eta^2}{(1 - \lambda^2)^3}\|Y_0 - \bar{Y}_0\|_F^2 + \sum_{t=0}^{T-1} \frac{16n\lambda^2 p_t(\eta^2 C_v^2 + r^2)}{(1 - \lambda^2)^2} \tag{14}
$$

where the last inequality comes from Eq. (11). When $\eta \leq \frac{(1-\lambda)^2}{40\lambda^2 L}$ we have $\frac{288L^2\eta^2\lambda^4(1+\lambda^2)}{(1-\lambda^2)^4} \leq \frac{1}{2}$ and

$$
\frac{1}{T}\sum_{t=1}^{T} \|X_t - \bar{X}_t\|_F^2
$$
$$
\leq \frac{192\eta^2\lambda^4(1 + \lambda^2)}{(1 - \lambda^2)^4 T}\sum_{t=0}^{T-1}\sum_{i=1}^{n} \|v_t^{(i)} - \nabla f_i(x_t^{(i)})\|^2 + \frac{288nL^2\eta^2\lambda^4(1 + \lambda^2)}{(1 - \lambda^2)^4 T}\sum_{t=0}^{T-1} \omega_t
$$

$$+ \frac{64\lambda^4\eta^2}{(1-\lambda^2)^3 T}\sum_{i=1}^{n}\|\nabla f_i(x_0)\|^2 + \frac{256\log(4/\delta)n\lambda^4\eta^2\sigma^2}{(1-\lambda^2)^3 b_0 T} + \sum_{t=0}^{T-1}\frac{32n\lambda^2 p_t(\eta^2 C_v^2 + r^2)}{(1-\lambda^2)^2 T}$$

$$\leq \frac{73728\log(4/\delta)L^2\eta^2\lambda^4(1+\lambda^2)}{(1-\lambda^2)^4 b_1\beta T}\sum_{t=0}^{T-1}\|X_t - \bar{X}_t\|_F^2 + \frac{24576n\log(4/\delta)L^2\eta^2\lambda^4(1+\lambda^2)}{(1-\lambda^2)^4 b_1\beta T}\sum_{t=0}^{T-1}\omega_t$$

$$+ \frac{n\log(4/\delta)\eta^2\lambda^4(1+\lambda^2)\sigma^2}{(1-\lambda^2)^4}\left(\frac{3072\beta}{b_1} + \frac{384}{\beta b_0 T}\right) + \frac{288nL^2\eta^2\lambda^4(1+\lambda^2)}{(1-\lambda^2)^4 T}\sum_{t=0}^{T-1}\omega_t$$

$$+ \frac{64\lambda^4\eta^2}{(1-\lambda^2)^3 T}\sum_{i=1}^{n}\|\nabla f_i(x_0)\|^2 + \frac{256\log(4/\delta)n\lambda^4\eta^2\sigma^2}{(1-\lambda^2)^3 b_0 T} + \sum_{t=0}^{T-1}\frac{32n\lambda^2 p_t(\eta^2 C_v^2 + r^2)}{(1-\lambda^2)^2 T} \tag{15}$$

where the last inequality is achieved by Lemma 1. According to the parameter setting, we have

$$\frac{73728\log(4/\delta)L^2\eta^2\lambda^4(1+\lambda^2)}{(1-\lambda^2)^4 b_1\beta} \leq \frac{1}{2}$$

Therefore, we have

$$\frac{1}{T}\sum_{t=1}^{T}\|X_t - \bar{X}_t\|_F^2$$

$$\leq \frac{160000n\log(4/\delta)L^2\eta^2\lambda^4}{(1-\lambda)^4\min\{b_1\beta,1\}T}\sum_{t=0}^{T-1}\omega_t + \frac{12288n\log(4/\delta)\beta\eta^2\lambda^4\sigma^2}{(1-\lambda)^4 b_1} + \frac{2000n\log(4/\delta)\eta^2\lambda^4\sigma^2}{(1-\lambda)^4\beta b_0 T}$$

$$+ \frac{128\lambda^4\eta^2}{(1-\lambda)^3 T}\sum_{i=1}^{n}\|\nabla f_i(x_0)\|^2 + \sum_{t=0}^{T-1}\frac{64n\lambda^2 p_t(\eta^2 C_v^2 + r^2)}{(1-\lambda)^2 T} \tag{16}$$

where we have used the condition $\lambda \leq 1$ to simplify the inequality. Moreover, sum Eq. (11) and we can achieve

$$\frac{1}{T}\sum_{t=0}^{T-1}\|Y_t - \bar{Y}_t\|_F^2$$

$$\leq \frac{12\lambda^2}{(1-\lambda)T}\sum_{t=0}^{T-1}\sum_{i=1}^{n}\|v_t^{(i)} - \nabla f_i(x_t^{(i)})\|^2 + \frac{36L^2\lambda^2}{(1-\lambda)T}\sum_{t=0}^{T-1}\|X_t - \bar{X}_t\|_F^2 + \frac{18nL^2\lambda^2}{(1-\lambda)T}\sum_{t=0}^{T-1}\omega_t$$

$$+ \frac{2}{(1-\lambda)T}\|Y_0 - \bar{Y}_0\|_F^2$$

$$\leq \frac{36L^2\lambda^2}{(1-\lambda)T}\left(1 + \frac{128\log(4/\delta)}{b_1\beta}\right)\sum_{t=0}^{T-1}\|X_t - \bar{X}_t\|_F^2 + \frac{18nL^2\lambda^2}{(1-\lambda)T}\left(1 + \frac{128\log(4/\delta)}{b_1\beta}\right)\sum_{t=0}^{T-1}\omega_t$$

$$+ \frac{192\log(4/\delta)n\lambda^2\beta\sigma^2}{(1-\lambda)b_1} + \frac{25\log(4/\delta)n\lambda^2\sigma^2}{(1-\lambda)\beta b_0 T} + \frac{4\lambda^2}{(1-\lambda)T}\sum_{i=1}^{n}\|\nabla f_i(x_0)\|^2$$

$$\leq \frac{4644\log(4/\delta)L^2\lambda^2}{(1-\lambda)\min\{b_1\beta,1\}T}\sum_{t=0}^{T-1}\|X_t - \bar{X}_t\|_F^2 + \frac{2322\log(4/\delta)nL^2\lambda^2}{(1-\lambda)\min\{b_1\beta,1\}T}\sum_{t=0}^{T-1}\omega_t$$

$$+ \frac{192\log(4/\delta)n\lambda^2\beta\sigma^2}{(1-\lambda)b_1} + \frac{25\log(4/\delta)n\lambda^2\sigma^2}{(1-\lambda)\beta b_0 T} + \frac{4\lambda^2}{(1-\lambda)T}\sum_{i=1}^{n}\|\nabla f_i(x_0)\|^2$$

$$\leq \frac{37152\log(4/\delta)L^2\eta^2\lambda^4}{(1-\lambda)^3\min\{b_1\beta,1\}T}\sum_{t=0}^{T-1}\|Y_t - \bar{Y}_t\|_F^2 + \frac{2322\log(4/\delta)nL^2\lambda^2}{(1-\lambda)\min\{b_1\beta,1\}T}\sum_{t=0}^{T-1}\omega_t$$

$$+ \frac{192\log(4/\delta)n\lambda^2\beta\sigma^2}{(1-\lambda)b_1} + \sum_{t=0}^{T-1}\frac{74304\log(4/\delta)nL^2\lambda^4 p_t(\eta^2 C_v^2 + r^2)}{(1-\lambda)^3\min\{b_1\beta,1\}T} + \frac{25\log(4/\delta)n\lambda^2\sigma^2}{(1-\lambda)\beta b_0 T}$$

$$+ \frac{4\lambda^2}{(1-\lambda)T}\sum_{i=1}^{n}\|\nabla f_i(x_0)\|^2 \tag{17}$$

where the second inequality uses Lemma 1 and Eq. (12). The last inequality uses the sum of Eq. (13). As $\frac{37152\log(4/\delta)L^2\eta^2\lambda^4}{(1-\lambda)^3\min\{b_1\beta,1\}} \leq \frac{1}{2}$, we have

$$
\frac{1}{T}\sum_{t=0}^{T-1}\|Y_t - \bar{Y}_t\|_F^2
$$

$$
\leq \frac{4644\log(4/\delta)nL^2\lambda^2}{(1-\lambda)\min\{b_1\beta,1\}T}\sum_{t=0}^{T-1}\omega_t + \frac{384\log(4/\delta)n\lambda^2\beta\sigma^2}{(1-\lambda)b_1} + \frac{50\log(4/\delta)n\lambda^2\sigma^2}{(1-\lambda)\beta b_0 T}
$$

$$
+ \frac{8\lambda^2}{(1-\lambda)T}\sum_{i=1}^{n}\|\nabla f_i(x_0)\|^2 + \sum_{t=0}^{T-1}\frac{150000\log(4/\delta)nL^2\lambda^4 p_t(\eta^2 C_v^2 + r^2)}{(1-\lambda)^3\min\{b_1\beta,1\}T} \tag{18}
$$

which finishes the proof. $\qquad\square$

### C.3   Proof of Lemma 3

*Proof.* By Assumption 3 we have

$$
f(\bar{x}_{t+1}) \leq f(\bar{x}_t) + \langle \nabla f(\bar{x}_t), \bar{x}_{t+1} - \bar{x}_t \rangle + \frac{L}{2}\|\bar{x}_{t+1} - \bar{x}_t\|^2
$$

$$
= f(\bar{x}_t) + \langle \nabla f(\bar{x}_t), -\eta\bar{v}_t \rangle + \langle \nabla f(\bar{x}_t), \bar{x}_{t+1} - \bar{x}_t + \eta\bar{v}_t \rangle + \frac{L}{2}\|\bar{x}_{t+1} - \bar{x}_t\|^2
$$

$$
= f(\bar{x}_t) - \frac{\eta}{2}\|\bar{v}_t\|^2 - \frac{\eta}{2}\|\nabla f(\bar{x}_t)\|^2 + \frac{\eta}{2}\|\bar{v}_t - \nabla f(\bar{x}_t)\|^2 + \frac{\eta}{2}\|\nabla f(\bar{x}_t)\|^2
$$

$$
+ \frac{1}{2\eta}\|\bar{x}_{t+1} - \bar{x}_t + \eta\bar{v}_t\|^2 - \frac{1}{2\eta}\|\bar{x}_{t+1} - \bar{x}_t + \eta\bar{v}_t - \eta\nabla f(\bar{x}_t)\|^2 + \frac{L}{2}\|\bar{x}_{t+1} - \bar{x}_t\|^2
$$

$$
\leq f(\bar{x}_t) - \frac{\eta}{2}\|\bar{v}_t\|^2 + \frac{\eta}{2}\|\bar{v}_t - \nabla f(\bar{x}_t)\|^2 + \frac{1}{2\eta}\|\bar{x}_{t+1} - \bar{x}_t + \eta\bar{v}_t\|^2 + \frac{L\omega_t}{2}
$$

$$
- \frac{1}{2\eta}\omega_t - \frac{\eta}{2}\|\bar{v}_t - \nabla f(\bar{x}_t)\|^2 + \frac{1}{4\eta}\omega_t + \eta\|\bar{v}_t - \nabla f(\bar{x}_t)\|^2
$$

$$
\leq f(\bar{x}_t) - \frac{1}{4\eta}\omega_t - \frac{\eta}{2}\|\bar{v}_t\|^2 + \frac{p_t(\eta^2 C_v^2 + r^2)}{\eta} + \frac{L\omega_t}{2} + 2\eta\|\bar{v}_t - \frac{1}{n}\sum_{i=1}^{n}\nabla f_i(x_t^{(i)})\|^2
$$

$$
+ \frac{L^2\eta}{n}\|X_t - \bar{X}_t\|_F^2 \tag{19}
$$

where the first inequality is obtained by Young's inequality and the last inequality is obtained by Cauchy-Schwartz inequality, Assumption 3 and the fact that perturbation is only drawn when $\|y_t^{(i)}\| \leq C_v$ and $n_t$ nodes draw perturbation in iteration $t$. Sum Eq. (19) and apply Lemma 1, we have

$$
f(\bar{x}_T) \leq f(x_0) - \frac{1}{4\eta}\sum_{t=0}^{T-1}\omega_t - \frac{\eta}{2}\sum_{t=0}^{T-1}\|\bar{v}_t\|^2 + \left(1 + \frac{768\log(4/\delta)}{nb_1\beta}\right)\frac{L^2\eta}{n}\sum_{t=0}^{T-1}\|X_t - \bar{X}_t\|_F^2
$$

$$
+ \sum_{t=0}^{T-1}\frac{p_t(\eta^2 C_v^2 + r^2)}{\eta} + \frac{32\log(4/\delta)\beta\eta T\sigma^2}{nb_1} + \left(1 + \frac{384\log(4/\delta)L\eta}{nb_1\beta}\right)\sum_{t=0}^{T-1}L\omega_t
$$

$$
+ \frac{4\log(4/\delta)\eta\sigma^2}{n\beta b_0} \tag{20}
$$

According to the update of gradient tracker, we have $\bar{y}_t = \bar{v}_t$. By Lemma 10 we have

$$
\frac{1}{n}\sum_{i=1}^{n}\|x_{t+1}^{(i)} - x_t^{(i)}\|^2 = \omega_t + \frac{1}{n}\|(X_{t+1} - \bar{X}_{t+1}) - (X_t - \bar{X}_t)\|_F^2 \tag{21}
$$

$$
\frac{1}{n}\sum_{i=1}^{n}\|y_t^{(i)}\|^2 = \|\bar{y}_t\|^2 + \frac{1}{n}\|Y_t - \bar{Y}_t\|_F^2 \tag{22}
$$

Divide the term $\|\bar{v}_t\|^2$ in Eq. (20) into three portions and we get

$$f(\bar{x}_T)$$

$$\leq f(x_0) - \frac{1}{8\eta} \sum_{t=0}^{T-1} \omega_t - \frac{(1-\lambda)^2}{256\eta} \sum_{t=0}^{T-1} \omega_t - \frac{\eta}{2} \sum_{t=0}^{T-1} \|\bar{v}_t\|^2 + (1 + \frac{768\log(4/\delta)}{nb_1\beta}) \frac{L^2\eta}{n} \sum_{t=0}^{T-1} \|X_t - \bar{X}_t\|_F^2$$

$$+ \sum_{t=0}^{T-1} \frac{p_t(\eta^2 C_v^2 + r^2)}{\eta} + \frac{32\log(4/\delta)\beta\eta T\sigma^2}{nb_1} + (1 + \frac{384\log(4/\delta)L\eta}{nb_1\beta}) \sum_{t=0}^{T-1} L\omega_t + \frac{4\log(4/\delta)\eta\sigma^2}{n\beta b_0}$$

$$\leq f(x_0) - \frac{1}{8\eta} \sum_{t=0}^{T-1} \omega_t - \frac{(1-\lambda)^2}{256\eta} \sum_{t=0}^{T-1} \omega_t - \frac{\eta}{2n} \sum_{t=0}^{T-1} \sum_{i=1}^{n} \|y_t^{(i)}\|^2 + \frac{\eta}{2n} \sum_{t=0}^{T-1} \|Y_t - \bar{Y}_t\|_F^2$$

$$+ \sum_{t=0}^{T-1} \frac{p_t(\eta^2 C_v^2 + r^2)}{\eta} + (1 + \frac{768\log(4/\delta)}{nb_1\beta}) \frac{L^2\eta}{n} \sum_{t=0}^{T-1} \|X_t - \bar{X}_t\|_F^2 + \frac{32\log(4/\delta)\beta\eta T\sigma^2}{nb_1}$$

$$+ (1 + \frac{384\log(4/\delta)L\eta}{nb_1\beta}) \sum_{t=0}^{T-1} L\omega_t + \frac{4\log(4/\delta)\eta\sigma^2}{n\beta b_0}$$

$$\leq f(x_0) - \frac{1}{8\eta} \sum_{t=0}^{T-1} \omega_t - \frac{(1-\lambda)^2}{256n\eta} \sum_{t=0}^{T-1} \sum_{i=1}^{n} \|x_{t+1}^{(i)} - x_t^{(i)}\|^2 - \frac{\eta}{2n} \sum_{t=0}^{T-1} \sum_{i=1}^{n} \|y_t^{(i)}\|^2$$

$$+ (1 + \frac{768\log(4/\delta)}{nb_1\beta} + \frac{(1-\lambda)^2}{128L^2\eta^2}) \frac{L^2\eta}{n} \sum_{t=0}^{T-1} \|X_t - \bar{X}_t\|_F^2 + \frac{\eta}{2n} \sum_{t=0}^{T-1} \|Y_t - \bar{Y}_t\|_F^2$$

$$+ (1 + \frac{384\log(4/\delta)L\eta}{nb_1\beta}) \sum_{t=0}^{T-1} L\omega_t + \sum_{t=0}^{T-1} \frac{p_t(\eta^2 C_v^2 + r^2)}{\eta} + \frac{32\log(4/\delta)\beta\eta T\sigma^2}{nb_1} + \frac{4\log(4/\delta)\eta\sigma^2}{n\beta b_0}$$

$$\leq f(x_0) - \frac{1}{8L\eta} \sum_{t=0}^{T-1} L\omega_t - \frac{(1-\lambda)^2}{256n\eta} \sum_{t=0}^{T-1} \sum_{i=1}^{n} \|x_{t+1}^{(i)} - x_t^{(i)}\|^2 - \frac{\eta}{2n} \sum_{t=0}^{T-1} \sum_{i=1}^{n} \|y_t^{(i)}\|^2$$

$$+ A_1 \sum_{t=0}^{T-1} L\omega_t + A_2 \frac{T\beta\eta\sigma^2}{b_1} + A_3 \frac{\eta\sigma^2}{\beta b_0} + A_4 \sum_{t=0}^{T-1} \frac{p_t(\eta^2 C_v^2 + r^2)}{\eta} + A_5 \frac{\eta}{n} \sum_{i=1}^{n} \|\nabla f_i(x_0)\|^2 \qquad (23)$$

In the second inequality we use Eq. (22). In the third inequality we use Eq. (21) and Cauchy-Schwartz inequality. In the last inequality we use Lemma 2 and the coefficients are

$$A_1 = 1 + \frac{384\log(4/\delta)L\eta}{nb_1\beta} + (1 + \frac{768\log(4/\delta)}{nb_1\beta} + \frac{(1-\lambda)^2}{128L^2\eta^2}) \frac{160000\log(4/\delta)L^3\eta^3\lambda^4}{(1-\lambda)^4 \min\{b_1\beta, 1\}} + \frac{774\log(4/\delta)L\eta\lambda^2}{(1-\lambda)}$$

$$A_2 = \frac{32\log(4/\delta)}{n} + (1 + \frac{768\log(4/\delta)}{nb_1\beta} + \frac{(1-\lambda)^2}{128L^2\eta^2}) \frac{12288\log(4/\delta)L^2\eta^2\lambda^4}{(1-\lambda)^4} + \frac{64\log(4/\delta)\lambda^2}{(1-\lambda)}$$

$$A_3 = \frac{4\log(4/\delta)}{n} + (1 + \frac{768\log(4/\delta)}{nb_1\beta} + \frac{(1-\lambda)^2}{128L^2\eta^2}) \frac{2000\log(4/\delta)L^2\eta^2\lambda^4}{(1-\lambda)^4} + \frac{10\log(4/\delta)\lambda^2}{(1-\lambda)}$$

$$A_4 = 1 + (1 + \frac{768\log(4/\delta)}{nb_1\beta} + \frac{(1-\lambda)^2}{128L^2\eta^2}) \frac{64\lambda^2 L^2\eta^2}{(1-\lambda)^2} + \frac{25000\log(4/\delta)L^2\eta^2\lambda^4}{(1-\lambda)^3}$$

$$A_5 = (1 + \frac{768\log(4/\delta)}{nb_1\beta} + \frac{(1-\lambda)^2}{128L^2\eta^2}) \frac{128\lambda^4 L^2\eta^2}{(1-\lambda)^3} + \frac{2\lambda^2}{1-\lambda}$$

According to the parameter setting, we have $A_1 \leq \frac{1}{16L\eta}$, $A_2 \leq \frac{200\log(4/\delta)}{(1-\lambda)^2}$, $A_3 \leq \frac{40\log(4/\delta)}{(1-\lambda)^2}$, $A_4 \leq \frac{7}{4}$ and $A_5 \leq \frac{5}{1-\lambda}$. Therefore, we have

$$f(\bar{x}_T) \leq f(x_0) + \frac{40\log(4/\delta)\eta\sigma^2}{(1-\lambda)^2\beta b_0} + \frac{5\eta}{(1-\lambda)n} \sum_{i=1}^{n} \|\nabla f_i(x_0)\|^2 - \sum_{t=0}^{T-1} \mathcal{D}_t$$

$$\leq f(x_0) + \frac{\sigma^2}{L} + \frac{1}{nL} \sum_{i=1}^{n} \|\nabla f_i(x_0)\|^2 - \sum_{t=0}^{T-1} \mathcal{D}_t \qquad (24)$$

where

$$\mathcal{D}_t = \frac{1}{16\eta} \omega_t + \frac{(1-\lambda)^2}{256n\eta} \sum_{i=1}^{n} \|x_{t+1}^{(i)} - x_t^{(i)}\|^2 + \frac{\eta}{2n} \sum_{i=1}^{n} \|y_t^{(i)}\|^2 - \frac{200\eta\epsilon^2\sigma^2}{(1-\lambda)^2 C_1^2} - \frac{7p_t(\eta^2 C_v^2 + r^2)}{4\eta} \qquad (25)$$

which reaches the conclusion. □

## C.4 Proof of Lemma 4

*Proof.* For convenience, the iteration that draws perturbation is considered to be included in the escaping phase. If an iteration belongs to type-A, *i.e.*, $p_t \geq \frac{1}{5}$, then at least $n/5$ worker nodes are in the escaping phase. If an iteration belongs to type-C, we have $\frac{1}{n}\sum_{i=1}^n \|y_t^{(i)}\|^2 \leq \frac{4C_v^2}{5}$. Therefore, there are at least $\frac{n}{5}$ worker nodes satisfying $\|y_t^{(i)}\| \leq C_v$, which also indicates that at least $\frac{n}{5}$ worker nodes are in the escaping phase. Then if iteration $t$ is either type-A or type-C, there must be $n/5$ worker nodes in the escaping phase. We denote the set of these $n/5$ worker nodes as $\mathcal{E}_t$. Furthermore, if this iteration $t$ is not one of the last $C_T$ iterations before termination, then there must exist $n/10$ worker nodes out of $\mathcal{E}_t$ such that they have not met the condition $esc^{(i)} \geq C_T$ and will break the escaping phase before meeting the condition because of the termination criterion in Algorithm 1. We use $\mathcal{B}_t$ to denote these worker nodes.

For each $i \in \mathcal{B}_t$, we have an interval $[a_t^{(i)}, b_t^{(i)}]$ such that $t \in [a_t^{(i)}, b_t^{(i)}]$ and node $i$ enters escaping phase in iteration $a_t^{(i)}$ and breaks escaping phase in iteration $b_t^{(i)}$. Besides, we also have

$$b_t^{(i)} - a_t^{(i)} \leq C_T \quad \text{and} \quad \|x_{b_t^{(i)}}^{(i)} - x_{a_t^{(i)}}^{(i)}\| \geq C_d$$

Then by Cauchy-Schwartz inequality we have

$$C_d^2 \leq \|x_{b_t^{(i)}}^{(i)} - x_{a_t^{(i)}}^{(i)}\|^2 \leq C_T \sum_{t=a_t^{(i)}}^{b_t^{(i)}} \|x_{t+1}^{(i)} - x_t^{(i)}\|^2 \tag{26}$$

Let $a_t = \min_i\{a_t^{(i)}\}$ and $b_t = \max_i\{b_t^{(i)}\}$. It is easy to check that $b_t - a_t \leq 2C_T$. Next, we will perform the refining step. If $t < t'$ are two iterations that are either type-A or type-C and $t' \in [a_t, b_t]$, then we make $a_{t'} = a_t$ and $b_{t'} = b_t$. Let $\mathcal{I} = \cup_t [a_t, b_t]$ for all type-A and type-C iterations $t$. Then $I$ can be written as disjoint union of

$$\mathcal{I} = \mathcal{I}_1 \cup \mathcal{I}_2 \cup \cdots \cup \mathcal{I}_k \tag{27}$$

because if $a_t \leq a_{t'} \leq b_t$ then $[a_t, b_t]$ and $[a_{t'}, b_{t'}]$ can be merged into one interval. Now we can see for each iteration $t$ that is either type-A or type-C and $t$ is not one of the last $C_T$ iterations, we have $t \in \mathcal{I}$. Next we will estimate the descent over $\mathcal{I}$. Without loss of generality, we consider an interval $\mathcal{I}_j$. $\mathcal{I}_j$ can be expressed by union $\mathcal{J}_1 \cup \cdots \cup \mathcal{J}_l$ where $\mathcal{J}_m = [a_{t_m}, b_{t_m}]$ for some $t_m$, $m = 1, \cdots, l$. Because of the refining step, we have each $t_m$ is only included in interval $\mathcal{J}_m$ and the intersection of any three intervals in $\mathcal{J}_1, \cdots, \mathcal{J}_l$ is $\emptyset$. According to Eq. (26) we have

$$\frac{1}{n}\sum_{i=1}^n \sum_{t \in \mathcal{J}_m} \|x_{t+1}^{(i)} - x_t^{(i)}\|^2 \geq \frac{C_d^2}{10C_T} \tag{28}$$

since $|\mathcal{B}_t| \geq \frac{n}{10}$. Next, we will consider the intersection of $\mathcal{J}_m$ and $\mathcal{J}_{m+1}$. Notice that when estimating Eq. (28) we only add the terms $\|x_{t+1}^{(i)} - x_t^{(i)}\|^2$ on nodes $i \in \mathcal{B}_{t_m}$ and in the intervals $[a_{t_m}^{(i)}, b_{t_m}^{(i)}]$. Therefore, for any node $i \notin \mathcal{B}_{t_m} \cap \mathcal{B}_{t_{m+1}}$, the terms used to estimate Eq. (28) will not be added repeatedly. If $i \in \mathcal{B}_{t_m} \cap \mathcal{B}_{t_{m+1}}$, we have $[a_{t_m}^{(i)}, b_{t_m}^{(i)}]$ and $[a_{t_{m+1}}^{(i)}, b_{t_{m+1}}^{(i)}]$ are disjoint because $t_{m+1} \in [a_{t_{m+1}}^{(i)}, b_{t_{m+1}}^{(i)}]$ but $t_{m+1} \notin [a_{t_m}^{(i)}, b_{t_m}^{(i)}]$ and a node cannot draw perturbation before breaking the escaping phase. Hence we can sum Eq. (28) over $m$ and achieve

$$\frac{1}{n}\sum_{i=1}^n \sum_{t \in \mathcal{I}_j} \|x_{t+1}^{(i)} - x_t^{(i)}\|^2 \geq \frac{lC_d^2}{10C_T} \tag{29}$$

Since the length of each $\mathcal{J}_m$ is not larger than $2C_T$, we have

$$\frac{1}{n}\sum_{i=1}^n \sum_{t \in \mathcal{I}_j} \|x_{t+1}^{(i)} - x_t^{(i)}\|^2 \geq \frac{|\mathcal{I}_j|C_d^2}{20C_T^2} \text{ and } \frac{1}{n}\sum_{i=1}^n \sum_{t \in \mathcal{I}} \|x_{t+1}^{(i)} - x_t^{(i)}\|^2 \geq \frac{|\mathcal{I}|C_d^2}{20C_T^2} \tag{30}$$

Combining Eq. (30) and Lemma 3, we can estimate the descent over $\mathcal{I}$ by

$$\sum_{t \in \mathcal{I}} \mathcal{D}_t \geq |\mathcal{I}|\Big(\frac{(1-\lambda)^2 C_d^2}{5120\eta C_T^2} - \frac{200\eta\epsilon^2\sigma^2}{(1-\lambda)^2 C_1^2} - \frac{7(\eta^2 C_v^2 + r^2)}{4\eta}\Big) \geq |\mathcal{I}| \cdot \frac{(1-\lambda)^2 C_2^2 \eta\epsilon^2}{10000} \tag{31}$$

according to the parameter setting. □

## C.5 Proof of Lemma 5

*Proof.* According to Lemma 3 and the definition of type-B iteration, we have

$$\mathcal{D}_t \geq \frac{\eta C_v^2}{20} - \frac{200\eta\epsilon^2}{(1-\lambda)^2 C_1^2} - \frac{7r^2}{20\eta} \geq \frac{\eta C_v^2}{40} - \frac{200\eta\epsilon^2\sigma^2}{(1-\lambda)^2 C_1^2} \geq \frac{(1-\lambda)^2 C_2^2 \eta\epsilon^2}{8000000} \tag{32}$$

for all type-B iteration $t$ where we have used the parameter setting. □

## C.6 Proof of Lemma 6

*Proof.* Suppose the conclusion is not true and we will find the conflict. Thus, we have the assumption that there are at least $\frac{n}{10}$ worker nodes satisfying $d_i \leq 2C_d$. First, we define

$$w_t^{(i)} = x_t^{(i)} - x_t^{(i)'}, \ w_t = \bar{x}_t - \bar{x}_t', \ \mathcal{H} = \nabla^2 f(\bar{x}_s), \ \mathcal{H}^{(i)} = \nabla^2 f_i(\bar{x}_s), \ \mathcal{H}_t^{(i)} = \nabla^2 F_i(x_s^{(i)}, \xi_t^{(i)})$$

$$\zeta_t = \frac{1}{n} \sum_{i=1}^n (\nabla F_i(x_t^{(i)}, \xi_t^{(i)}) - \nabla F_i(\bar{x}_t, \xi_t^{(i)})) - (\nabla F_i(x_t^{(i)'}, \xi_t^{(i)}) - \nabla F_i(\bar{x}_t', \xi_t^{(i)}))$$

$$- (1-\beta)(\nabla F_i(x_{t-1}^{(i)}, \xi_t^{(i)}) - \nabla F_i(\bar{x}_{t-1}, \xi_t^{(i)})) - (\nabla F_i(x_{t-1}^{(i)'}, \xi_t^{(i)}) - \nabla F_i(\bar{x}_{t-1}', \xi_t^{(i)}))$$

$$\nu_t = \bar{v}_t - \nabla f(\bar{x}_t) - (\bar{v}_t' - \nabla f(\bar{x}_t')) - \zeta_t$$

and

$$\bar{\Delta}_t = \int_0^1 (\nabla^2 f(\bar{x}_t' + \theta(\bar{x}_t - \bar{x}_t')) - \mathcal{H})d\theta$$

$$\Delta_t^{(i)} = \int_0^1 (\nabla^2 f_i(\bar{x}_t' + \theta(\bar{x}_t - \bar{x}_t')) - \mathcal{H}^{(i)})d\theta$$

Then we have

$$w_t = w_{t-1} - \eta(\bar{v}_{t-1} - \bar{v}_{t-1}')$$

$$= w_{t-1} - \eta(\nabla f(\bar{x}_{t-1}) - \nabla f(\bar{x}_{t-1}') + \bar{v}_{t-1} - \nabla f(\bar{x}_{t-1}) - \bar{v}_{t-1}' + \nabla f(\bar{x}_{t-1}'))$$

$$= w_{t-1} - \eta\Big[(\bar{x}_{t-1} - \bar{x}_{t-1}') \int_0^1 \nabla^2 f(\bar{x}_{t-1}' + \theta(\bar{x}_{t-1} - \bar{x}_{t-1}'))d\theta + \nu_{t-1} + \zeta_{t-1}\Big]$$

$$= (I - \eta\mathcal{H})w_{t-1} - \eta(\bar{\Delta}_{t-1}w_{t-1} + \nu_{t-1} + \zeta_{t-1}) \tag{33}$$

Here term $\zeta_t$ is yield from consensus error and does not exist in centralized algorithms. Applying recursion to Eq. (33), we can obtain

$$w_t = (I - \eta\mathcal{H})^{t-s-1} w_{s+1} - \eta \sum_{\tau=s+1}^{t-1} (I - \eta\mathcal{H})^{t-\tau-1}(\bar{\Delta}_\tau w_\tau + \nu_\tau + \zeta_\tau) \tag{34}$$

Let $q_t = \eta \sum_{\tau=s+1}^{t-1} (I - \eta\mathcal{H})^{t-\tau-1}(\bar{\Delta}_\tau w_\tau + \nu_\tau + \zeta_\tau)$. We will prove

$$\|q_t\| \leq \frac{1}{2}(1 + \eta\gamma)^{t-s-1} p_s r_0 \tag{35}$$

which leads to

$$\frac{1}{2}(1 + \eta\gamma)^{t-s-1} p_s r_0 \leq \|w_t\| \leq \frac{3}{2}(1 + \eta\gamma)^{t-s-1} p_s r_0 \tag{36}$$

because $\|(I-\eta\mathcal{H})^{t-s-1} w_{s+1}\| = (1+\eta\gamma)^{t-s-1} p_s r_0$ according to the definition of $w_{s+1}$. We define $\bar{d} = \max_{s \leq t \leq s+C_T}\{\|\bar{x}_t - \bar{x}_s\|, \|\bar{x}_t' - \bar{x}_s\|\}$. Since at least $\frac{n}{10}$ nodes satisfy $d_i \leq 2C_d$, $C_d = \tilde{O}(\epsilon^{1-\alpha})$ and the averaged consensus error is bounded by $O(\epsilon^{2(1+\theta)})$, we have

$$d_i \leq 3C_d \quad \text{and} \quad \bar{d} \leq \frac{1}{n}\sum_{i=1}^n d_i \leq 3C_d \tag{37}$$

To achieve Eq. (35), it is sufficient to prove

$$\eta \sum_{\tau=s+1}^{t-1} (1+\eta\gamma)^{t-\tau-1}\|\bar{\Delta}_\tau w_\tau\| + \|\nu_\tau\| + \|\zeta_\tau\| \leq \frac{1}{2}(1+\eta\gamma)^{t-s-1}p_s r_0 \tag{38}$$

$$\|\nu_t\| \leq \sqrt{\frac{4\log(4/\delta)}{b_1}} \cdot \frac{(1+\eta\gamma)^{t-s-1}Lp_s r_0}{t-s} + \frac{1}{12\eta C_T}(1+\eta\gamma)^{t-s-1}p_s r_0 \tag{39}$$

$$\|\zeta_t\| \leq 8\left(\frac{1+\lambda^2}{2}\right)^{\frac{t-s-1}{2}}L\sqrt{p_s}r_0 + \frac{L\eta(1+\eta\gamma)^{t-s-1}Lp_s r_0}{\sqrt{b_1}(t-s)} + \frac{1}{12\eta C_T}(1+\eta\gamma)^{t-s-1}p_s r_0 \tag{40}$$

which can be derived by induction. When $t = s + 1$, the left side of Eq. (38) is 0 and thus the inequality is satisfied. Suppose Eq. (38) holds for $t \leq t_0$. When $t = t_0 + 1$, we have

$$\eta \sum_{\tau=s+1}^{t-1} (1+\eta\gamma)^{t-\tau-1}\|\bar{\Delta}_\tau w_\tau\|$$

$$\leq \frac{3}{2}\eta\rho\bar{d} \sum_{\tau=s+1}^{t-1} (1+\eta\gamma)^{t-s-2}p_s r_0 \leq 5\eta\rho C_d C_T(1+\eta\gamma)^{t-s-2}p_s r_0$$

$$\leq \frac{1}{6}(1+\eta\gamma)^{t-s-1}p_s r_0 \tag{41}$$

where we use Assumption 4 and the case of $t \leq t_0$ in the first two inequalities. We use the parameter setting of $C_d$ in the last inequality. Next, we will estimate the terms related to $\nu_t$. By Azuma-Hoeffding inequality we know Eq. (39) is satisfied when $t = s + 1$. We define

$$\epsilon_{t,i} = (\nabla F_i(\bar{x}_{t+1}, \xi_{t+1}^{(i)}) - \nabla f_i(\bar{x}_{t+1})) - (1-\beta)(\nabla F_i(\bar{x}_t, \xi_{t+1}) - \nabla f_i(\bar{x}_t))$$

$$\epsilon'_{t,i} = (\nabla F_i(\bar{x}'_{t+1}, \xi_{t+1}^{(i)}) - \nabla f_i(\bar{x}'_{t+1})) - (1-\beta)(\nabla F_i(\bar{x}'_t, \xi_{t+1}) - \nabla f_i(\bar{x}'_t))$$

Then according to the definition of $\nu_t$ we have

$$\nu_{t+1} = (1-\beta)\nu_t + \frac{1}{n}\sum_{i=1}^{n}(\epsilon_{t,i} - \epsilon'_{t,i}) = \frac{1}{n}\sum_{\tau=s}^{t}(1-\beta)^{t-\tau}\sum_{i=1}^{n}(\epsilon_{\tau,i} - \epsilon'_{\tau,i}) \tag{42}$$

Define

$$\tilde{\Delta}_{t,1}^{(i)} = \int_0^1 (\nabla^2 F_i(\bar{x}'_t + \theta(\bar{x}_t - \bar{x}'_t), \xi_t^{(i)}) - \mathcal{H}_t^{(i)})d\theta$$

$$\tilde{\Delta}_{t,2}^{(i)} = \int_0^1 (\nabla^2 F_i(\bar{x}'_{t-1} + \theta(\bar{x}_{t-1} - \bar{x}'_{t-1}), \xi_t^{(i)}) - \mathcal{H}_t^{(i)})d\theta$$

$$\hat{\Delta}_{t,1}^{(i)} = \int_0^1 (\nabla^2 F_i(x_t^{(i)'} + \theta(x_t^{(i)} - x_t^{(i)'}), \xi_t^{(i)}) - \mathcal{H}_t^{(i)})d\theta$$

$$\hat{\Delta}_{t,2}^{(i)} = \int_0^1 (\nabla^2 F_i(x_{t-1}^{(i)'} + \theta(x_{t-1}^{(i)} - x_{t-1}^{(i)'}), \xi_t^{(i)}) - \mathcal{H}_t^{(i)})d\theta \tag{43}$$

Then we have

$$\epsilon_{t,i} - \epsilon'_{t,i}$$
$$= \mathcal{H}_{t+1}^{(i)}w_{t+1} + \tilde{\Delta}_{t+1,1}^{(i)}w_{t+1} - \mathcal{H}^{(i)}w_{t+1} - \Delta_{t+1}^{(i)}w_{t+1} + (1-\beta)(\mathcal{H}^{(i)}w_t + \Delta_t^{(i)}w_t)$$
$$\quad - (1-\beta)(\mathcal{H}_{t+1}^{(i)}w_t + \tilde{\Delta}_{t+1,2}^{(i)}w_t)$$
$$= (\mathcal{H}_{t+1}^{(i)} - \mathcal{H})(w_{t+1} - (1-\beta)w_t) + (\tilde{\Delta}_{t+1,1}^{(i)} - \Delta_{t+1}^{(i)})w_{t+1} + (1-\beta)(\Delta_t^{(i)} - \tilde{\Delta}_{t+1,2}^{(i)})w_t \tag{44}$$

According to Assumption 3 and Assumption 4, we have

$$\|\epsilon_{t,i} - \epsilon'_{t,i}\| \leq 2L\|w_{t+1} - w_t\| + (2\beta L + 3\rho C_d)\|w_t\| + 3\rho C_d\|w_{t+1}\| \tag{45}$$

Applying Azuma-Hoeffding inequality to Eq. (42), with Eq. (45) we can obtain

$$\|\nu_t\|^2 \leq \frac{4\log(4/\delta)}{nb_1} \sum_{\tau=s}^{t-1} [2L\|w_{\tau+1} - w_\tau\| + (2\beta L + 3\rho C_d)\|w_\tau\| + 3\rho C_d\|w_{\tau+1}\|]^2$$

$$\leq \frac{48\log(4/\delta)}{nb_1} \sum_{\tau=s+1}^{t} (L^2\|w_\tau - w_{\tau-1}\|^2 + 5\rho^2 C_d^2\|w_\tau\|^2) \tag{46}$$

since $\beta$ is $\Theta(\epsilon^{1+\theta})$ and $C_d$ is $\Theta(\epsilon^{1-\alpha})$. According to Eq. (34), we have

$$L\|w_\tau - w_{\tau-1}\|$$

$$= L\| - \eta\mathcal{H}(I - \eta\mathcal{H})^{\tau-s-2}w_{s+1} - \eta \sum_{\tau'=s+1}^{\tau-2} \eta\mathcal{H}(I - \eta\mathcal{H})^{\tau'-s-2}(\bar{\Delta}_{\tau'}w_{\tau'} + \nu_{\tau'} + \zeta_{\tau'})$$

$$+ \eta(\bar{\Delta}_{\tau-1}w_{\tau-1} + \nu_{\tau-1} + \zeta_{\tau-1})\|$$

$$\leq L\eta\gamma(1 + \eta\gamma)^{\tau-s-2}p_s r_0 + \frac{L\eta\gamma}{2}(1 + \eta\gamma)^{\tau-s-2}p_s r_0 + L\eta\|\bar{\Delta}_{\tau-1}w_{\tau-1} + \nu_{\tau-1} + \zeta_{\tau-1}\|$$

$$\leq 2L\eta\gamma(1 + \eta\gamma)^{\tau-s-2}p_s r_0 + L\eta\|\bar{\Delta}_{\tau-1}w_{\tau-1} + \nu_{\tau-1} + \zeta_{\tau-1}\| \tag{47}$$

In the first inequality, the first term is derived by the definition of $w_{s+1}$. The second term is derived by the supposition that Eq. (38) holds for $t \leq t_0$ and the fact that Eq. (38) implies

$$\eta \sum_{\tau=s+1}^{t-1} (1 + \eta\gamma)^{t-\tau-1}\|\bar{\Delta}_\tau w_\tau + \nu_\tau + \zeta_\tau\| \leq \frac{1}{2}(1 + \eta\gamma)^{t-s-1}p_s r_0 \tag{48}$$

Combining Eq. (46) and Eq. (47), we have

$$\|\nu_t\|^2 \leq \frac{48\log(4/\delta)}{nb_1} \sum_{\tau=s+1}^{t} (L^2\|w_\tau - w_{\tau-1}\|^2 + 5\rho^2 C_d^2\|w_\tau\|^2)$$

$$\leq \frac{270\log(4/\delta)\rho^2 C_d^2}{nb_1\eta\gamma}(1 + \eta\gamma)^{2(t-s-1)}p_s^2 r_0^2 + \frac{192\log(4/\delta)L^2\eta\gamma}{nb_1}(1 + \eta\gamma)^{2(t-s-1)}p_s^2 r_0^2$$

$$+ \frac{96\log(4/\delta)L^2\eta^2}{nb_1} \sum_{\tau=s+1}^{t-2} \|\bar{\Delta}_\tau w_\tau + \nu_\tau + \zeta_\tau\|^2$$

$$\leq \frac{300\log(4/\delta)\rho^2 C_d^2}{nb_1\eta\gamma}(1 + \eta\gamma)^{2(t-s-1)}p_s^2 r_0^2 + \frac{192\log(4/\delta)L^2\eta\gamma}{nb_1}(1 + \eta\gamma)^{2(t-s-1)}p_s^2 r_0^2$$

$$+ \frac{4\log(4/\delta)L^2}{nb_1\eta\gamma C_T^2}(1 + \eta\gamma)^{2(t-s-1)}p_s^2 r_0^2 + \frac{5000\log^2(4/\delta)L^4\eta^2 p_s^2 r_0^2}{b_1^2} \sum_{\tau=s+1}^{t-2} \frac{(1 + \eta\gamma)^{2(\tau-s-1)}}{(\tau - s)^2}$$

$$\leq \frac{1}{288\eta^2 C_T^2}(1 + \eta\gamma)^{2(t-s-1)}p_s^2 r_0^2 + \frac{800\log^2(4/\delta)L^2\eta\gamma}{nb_1}(1 + \eta\gamma)^{2(t-s-1)}p_s^2 r_0^2$$

$$+ \frac{5000\log^2(4/\delta)L^4\eta^2 p_s^2 r_0^2}{b_1^2} \sum_{\tau=s+1}^{s+\frac{L\eta}{b_1\gamma}} \frac{(1 + \eta\gamma)^{2(\tau-s-1)}}{(\tau - s)^2}$$

$$\leq \frac{10000\log(4/\delta)L^4\eta^4}{b_1^4\gamma^2} \cdot \frac{(1 + \eta\gamma)^{2(t-s-1)}L^2 p_s^2 r_0^2}{(t - s)^2} + \frac{1}{144\eta^2 C_T^2}(1 + \eta\gamma)^{2(t-s-1)}p_s^2 r_0^2$$

$$\leq \frac{(1 + \eta\gamma)^{2(t-s-1)}L^2 p_s^2 r_0^2}{(t - s)^2} + \frac{1}{144\eta^2 C_T^2}(1 + \eta\gamma)^{2(t-s-1)}p_s^2 r_0^2 \tag{49}$$

The exponential term in Eq. (40) can be addressed by the following strategy. When $t \geq \tilde{O}(\frac{1}{1-\lambda})$, the term can be dominated by other terms such as $\frac{1}{\eta C_T}$. When $t < \tilde{O}(\frac{1}{1-\lambda})$, it can be bounded by

$$\frac{L^2\eta^2\log(4/\delta)p_s r_0^2}{n(1-\lambda)b_1} \leq \frac{L^2\eta^2(t-s)^2\log(4/\delta)p_s^2 r_0^2}{(1-\lambda)b_1(t-s)^2} \tag{50}$$

The term $t - s$ in the numerator will be bounded by $\eta$ in this case and hence it can be merged to the first term in Eq. (39). In the third inequality of Eq. (49), we split the last term into two parts: $\tau - s > \frac{L\eta}{b_1\gamma}$ and $\tau - s \leq \frac{L\eta}{b_1\gamma}$. Since $\int_t^{+\infty} \frac{dx}{x^2} = \frac{1}{t}$, we can merge the case $\tau - s > \frac{L\eta}{b_1\gamma}$ into the second term and estimate the rest one where $\tau - s$ is small. According to the choice of $\theta$, we have $b_1 \geq \Theta(\epsilon^{2-\theta-5\alpha})$ and $\frac{\eta^2 C_d^2 C_T^3}{b_1} \leq O(1)$ and hence get the estimation in Eq. (49). We should notice that we use the relation $\frac{\eta}{b_1\gamma} \leq O(1)$ in our proof, which is automatically satisfied. By Eq. (49) we can reach the conclusion in Eq. (39). Furthermore, we have

$$\eta \sum_{\tau=s+1}^{t-1} (1+\eta\gamma)^{t-\tau-1} \|\nu_\tau\|$$

$$\leq L\eta(1+\eta\gamma)^{t-s-1} p_s r_0 \left( \sum_{\tau=s+1}^{t-1} \frac{1}{\tau-s} \right) + \frac{1}{12}(1+\eta\gamma)^{t-s-1} p_s r_0$$

$$\leq L\eta \log(C_T)(1+\eta\gamma)^{t-s-1} p_s r_0 + \frac{1}{12}(1+\eta\gamma)^{t-s-1} p_s r_0 \leq \frac{1}{6}(1+\eta\gamma)^{t-s-1} p_s r_0 \qquad (51)$$

The last step to prove Eq. (38) is to estimate the term corresponding to $\zeta_t$, which is a new term only occurred in decentralized algorithms. Recall the definitions in Eq. (43), we have

$$\zeta_t = \frac{1}{n} \sum_{i=1}^n \left[ (\mathcal{H}_t^{(i)} + \hat{\Delta}_{t,1}^{(i)})w_t^{(i)} - (\mathcal{H}_t^{(i)} + \tilde{\Delta}_{t,1}^{(i)})w_t - (1-\beta)((\mathcal{H}_t^{(i)} + \hat{\Delta}_{t,2}^{(i)})w_{t-1}^{(i)} - (\mathcal{H}_t^{(i)} + \tilde{\Delta}_{t,2}^{(i)})w_{t-1}) \right]$$

$$= \frac{1}{n} \sum_{i=1}^n \mathcal{H}_t^{(i)}[(w_t^{(i)} - w_t) - (1-\beta)(w_{t-1}^{(i)} - w_{t-1})] + \frac{1}{n} \sum_{i=1}^n \hat{\Delta}_{t,1}^{(i)}(w_t^{(i)} - w_t)$$

$$+ \frac{1}{n} \sum_{i=1}^n (\hat{\Delta}_{t,1}^{(i)} - \tilde{\Delta}_{t,1}^{(i)})w_t - \frac{1-\beta}{n} \sum_{i=1}^n \hat{\Delta}_{t,2}^{(i)}(w_{t-1}^{(i)} - w_{t-1}) - \frac{1-\beta}{n} \sum_{i=1}^n (\hat{\Delta}_{t,2}^{(i)} - \tilde{\Delta}_{t,2}^{(i)})w_{t-1} \qquad (52)$$

Then by Assumption 3, Assumption 4, Eq. (37), Lemma 10 and Cauchy-Schwartz inequality, we have

$$\|\zeta_t\|^2 \leq \frac{4L^2}{n}(\|X_t - \bar{X}_t - (X_t' - \bar{X}_t')\|_F^2 + \|X_{t-1} - \bar{X}_{t-1} - (X_{t-1}' - \bar{X}_{t-1}')\|_F^2)$$
$$+ 144\rho^2 C_d^2(\|w_t\|^2 + \|w_{t-1}\|^2) \qquad (53)$$

It is sufficient to prove

$$\frac{L}{\sqrt{n}}\|X_t - \bar{X}_t - (X_t' - \bar{X}_t')\|_F \leq 2\left(\frac{1+\lambda^2}{2}\right)^{\frac{t-s-1}{2}} L\sqrt{p_s}r_0 + \frac{1}{48\eta C_T}(1+\eta\gamma)^{t-s-1} p_s r_0$$

$$+ \frac{L\eta(1+\eta\gamma)^{t-s-1} L p_s r_0}{4\sqrt{b_1}(t-s)} \qquad (54)$$

because of Eq. (53) and the parameter setting. Eq. (54) can also be proven by induction. When $t = s+1$ the condition is satisfied. Next we will estimate $\|X_t - \bar{X}_t - (X_t' - \bar{X}_t')\|_F^2$. By Assumption 5 and Young's inequality we have

$$\|X_t - \bar{X}_t - (X_t' - \bar{X}_t')\|_F^2$$
$$= \|(W-J)[(X_{t-1} - \bar{X}_{t-1} - (X_{t-1}' - \bar{X}_{t-1}')) - \eta(Y_{t-1} - \bar{Y}_{t-1} - (Y_{t-1}' - \bar{Y}_{t-1}'))]\|_F^2$$
$$\leq \frac{1+\lambda^2}{2}\|X_{t-1} - \bar{X}_{t-1} - (X_{t-1}' - \bar{X}_{t-1}')\|_F^2 + \frac{2\eta^2\lambda^2}{1-\lambda^2}\|Y_{t-1} - \bar{Y}_{t-1} - (Y_{t-1}' - \bar{Y}_{t-1}')\|_F^2$$
$$\leq \frac{2\eta^2\lambda^2}{1-\lambda^2} \sum_{\tau=s+1}^{t-1} \left(\frac{1+\lambda^2}{2}\right)^{t-\tau-1}\|Y_\tau - \bar{Y}_\tau - (Y_\tau' - \bar{Y}_\tau')\|_F^2$$
$$+ \left(\frac{1+\lambda^2}{2}\right)^{t-s-1}\|X_{s+1} - \bar{X}_{s+1} - (X_{s+1}' - \bar{X}_{s+1}')\|_F^2$$
$$= \frac{2\eta^2\lambda^2}{1-\lambda} \sum_{\tau=s+1}^{t-1} \left(\frac{1+\lambda^2}{2}\right)^{t-\tau-1}\|Y_\tau - \bar{Y}_\tau - (Y_\tau' - \bar{Y}_\tau')\|_F^2 + \left(\frac{1+\lambda^2}{2}\right)^{t-s-1}\lambda^2(n-n_s)p_s r_0^2 \qquad (55)$$

where we apply recursion in the second inequality and use the definition of the decoupled sequences in the last equality. Similarly, by recursion we also have

$$
\|Y_t - \bar{Y}_t - (Y'_t - \bar{Y}'_t)\|_F^2
$$
$$
\leq \frac{1+\lambda^2}{2}\|Y_{t-1} - \bar{Y}_{t-1} - (Y'_{t-1} - \bar{Y}'_{t-1})\|_F^2 + \frac{\lambda^2+\lambda^4}{1-\lambda^2}\|V_t - V_{t-1} - (V'_t - V'_{t-1})\|_F^2
$$
$$
\leq \frac{2\lambda^2}{1-\lambda}\sum_{\tau=s+1}^{t}(\frac{1+\lambda^2}{2})^{t-\tau}\|V_\tau - V_{\tau-1} - (V'_\tau - V'_{\tau-1})\|_F^2 \tag{56}
$$

Combining above two inequalities, we achieve

$$
\|X_t - \bar{X}_t - (X'_t - \bar{X}'_t)\|_F^2
$$
$$
\leq \frac{2\eta^2\lambda^4}{(1-\lambda)^2}\sum_{\tau=s+1}^{t-1}(\frac{1+\lambda^2}{2})^{t-\tau-1}(t-\tau)\|V_\tau - V_{\tau-1} - (V'_\tau - V'_{\tau-1})\|_F^2
$$
$$
+ (\frac{1+\lambda^2}{2})^{t-s-1}\lambda^2(n-n_s)p_s r_0^2 \tag{57}
$$

According to the update rule of $v_t^{(i)}$ we have

$$
v_t^{(i)} - v_{t-1}^{(i)} - (v_t^{(i)'} - v_{t-1}^{(i)'}) - (1-\beta)(v_{t-1}^{(i)} - v_{t-2}^{(i)} - (v_{t-1}^{(i)'} - v_{t-2}^{(i)'}))
$$
$$
= \nabla F_i(x_t^{(i)}, \xi_t^{(i)}) - (1-\beta)\nabla F_i(x_{t-1}^{(i)}, \xi_t^{(i)}) - \nabla F_i(x_t^{(i)'}, \xi_t^{(i)}) + (1-\beta)\nabla F_i(x_{t-1}^{(i)'}, \xi_t^{(i)})
$$
$$
- [\nabla F_i(x_{t-1}^{(i)}, \xi_{t-1}^{(i)}) - (1-\beta)\nabla F_i(x_{t-2}^{(i)}, \xi_{t-1}^{(i)}) - \nabla F_i(x_{t-1}^{(i)'}, \xi_{t-1}^{(i)})
$$
$$
+ (1-\beta)\nabla F_i(x_{t-2}^{(i)'}, \xi_{t-1}^{(i)})] \tag{58}
$$

Then mimic the estimation of $\nu_t$, we can obtain

$$
\|V_t - V_{t-1} - (V'_t - V'_{t-1})\|_F^2
$$
$$
\leq \frac{32\log(4/\delta)}{b_1}\sum_{\tau=s}^{t-1}\sum_{i=1}^{n}[2L\|w_{\tau+1}^{(i)} - w_\tau^{(i)}\| + (2\beta L + 3\rho C_d)\|w_\tau^{(i)}\| + 3\rho C_d\|w_{\tau+1}^{(i)}\|]^2
$$
$$
+ 4L^2\sum_{i=1}^{n}\|w_t^{(i)} - w_{t-1}^{(i)}\|^2 + 36\rho^2 C_d^2\sum_{i=1}^{n}\|w_t^{(i)}\|^2 \tag{59}
$$

Combining above inequalities and the parameter setting of $\beta$, we can obtain

$$
\frac{1}{n}\|X_t - \bar{X}_t - (X'_t - \bar{X}'_t)\|_F^2 - (\frac{1+\lambda^2}{2})^{t-s-1}\lambda^2 p_s r_0^2
$$
$$
\leq (\frac{2000\log(4/\delta)\eta^2\rho^2 C_d^2(t-s)\lambda^4}{(1-\lambda)^2 b_1} + \frac{72\eta^2\rho^2 C_d^2\lambda^4}{(1-\lambda)^2})\sum_{\tau=s+1}^{t-1}(\frac{1+\lambda^2}{2})^{t-\tau-1}\frac{(t-\tau)}{n}\sum_{i=1}^{n}\|w_\tau^{(i)}\|^2
$$
$$
+ (\frac{500\log(4/\delta)L^2\eta^2(t-s)\lambda^4}{(1-\lambda)^2 b_1} + \frac{8L^2\eta^2\lambda^4}{(1-\lambda)^2})\sum_{\tau=s+1}^{t-1}(\frac{1+\lambda^2}{2})^{t-\tau-1}\frac{(t-\tau)}{n}\sum_{i=1}^{n}\|w_\tau^{(i)} - w_{\tau-1}^{(i)}\|^2
$$
$$
\leq \frac{L^2\eta^2\lambda^4}{(1-\lambda)^2}(32 + \frac{2000\log(4/\delta)(t-s)}{b_1})\sum_{\tau=s+1}^{t-1}(\frac{1+\lambda^2}{2})^{t-\tau-1}\frac{(t-\tau)}{n}\sum_{i=1}^{n}\|X_\tau - \bar{X}_\tau - (X'_\tau - \bar{X}'_\tau)\|_F^2
$$
$$
+ (\frac{2000\log(4/\delta)\eta^2\rho^2 C_d^2(t-s)\lambda^4}{(1-\lambda)^2 b_1} + \frac{72\eta^2\rho^2 C_d^2\lambda^4}{(1-\lambda)^2})\sum_{\tau=s+1}^{t-1}(\frac{1+\lambda^2}{2})^{t-\tau-1}(t-\tau)\|w_\tau\|^2
$$
$$
+ (\frac{500\log(4/\delta)L^2\eta^2(t-s)\lambda^4}{(1-\lambda)^2 b_1} + \frac{8L^2\eta^2\lambda^4}{(1-\lambda)^2})\sum_{\tau=s+1}^{t-1}(\frac{1+\lambda^2}{2})^{t-\tau-1}(t-\tau)\|w_\tau - w_{\tau-1}\|^2 \tag{60}
$$

Using Eq. (36), Eq. (47) and Eq. (54) we have

$$
\frac{L^2}{n}\|X_t - \bar{X}_t - (X'_t - \bar{X}'_t)\|_F^2
$$

$$\leq B_1 t^2 \left(\frac{1+\lambda^2}{2}\right)^{t-s-1} L^2 p_s r_0^2 + B_2 (1+\eta\gamma)^{2(t-s-1)} L^2 p_s^2 r_0^2 \sum_{\tau=s+1}^{t-1} \left(\frac{(1+\lambda^2)(1+\eta\gamma)^2}{2}\right)^{t-\tau-1} (t-\tau)$$

$$+ B_3 (1+\eta\gamma)^{2(t-s-1)} L^2 p_s^2 r_0^2 \sum_{\tau=s+1}^{t-1} \left(\frac{(1+\lambda^2)(1+\eta\gamma)^2}{2}\right)^{t-\tau-1} \frac{t-\tau}{(\tau-s)^2}$$

$$+ \left(\frac{1+\lambda^2}{2}\right)^{t-s-1} L^2 p_s r_0^2 \tag{61}$$

where

$$B_1 = \frac{4L^2\eta^2}{(1-\lambda)^2}\left(32 + \frac{2000\log(4/\delta)(t-s)}{b_1}\right) + 384 L^2 \eta^2 \left(\frac{500\log(4/\delta)L^2\eta^2(t-s)}{(1-\lambda)^2 b_1} + \frac{8L^2\eta^2}{(1-\lambda)^2}\right)$$

$$B_2 = \frac{1}{72(1-\lambda)^2 C_T^2}\left(2 + \frac{125\log(4/\delta)(t-s)}{b_1}\right) + \frac{4500\log(4/\delta)\eta^2\rho^2 C_d^2 (t-s)}{(1-\lambda)^2 b_1} + \frac{162\eta^2\rho^2 C_d^2}{(1-\lambda)^2}$$

$$+ \left(8L^2\eta^2\gamma^2 + 54 L^2\eta^2\rho^2 C_d^2 + \frac{1}{6C_T^2}\right)\left(\frac{500\log(4/\delta)L^2\eta^2(t-s)}{(1-\lambda)^2 b_1} + \frac{8L^2\eta^2}{(1-\lambda)^2}\right)$$

$$B_3 = \frac{48\log(4/\delta)L^2\eta^2}{b_1}\left(\frac{500\log(4/\delta)L^2\eta^2(t-s)}{(1-\lambda)^2 b_1} + \frac{8L^2\eta^2}{(1-\lambda)^2}\right)$$

$$+ \frac{L^4\eta^4(1+\eta\gamma)^{2(t-s-1)}}{(1-\lambda)^2 b_1}\left(32 + \frac{2000\log(4/\delta)(t-s)}{b_1}\right) \tag{62}$$

If $t \geq \tilde{O}(\frac{1}{1-\lambda})$, $t^2(\frac{1+\lambda^2}{2})^{t-s-1}$ is small and the first term of Eq. (61) can be merged to the second term. Otherwise if $t < \tilde{O}(\frac{1}{1-\lambda})$, it can be merged to the last term according to the parameter setting of $\eta$ and $b_1$. When $\epsilon$ is small, we have $\frac{(1+\lambda^2)(1+\eta\gamma)^2}{2} \leq \frac{3+\lambda^2}{4}$. Hence the second term of Eq. (61) can be bounded by Lemma 8. The third term of Eq. (61) can be estimated by Lemma 9 (the case of $t < \tilde{O}(\frac{1}{1-\lambda})$ can be addressed by the parameter setting of $\eta$ and $b_1$). Therefore, we can prove

$$\frac{L^2}{n}\|X_t - \bar{X}_t - (X'_t - \bar{X}'_t)\|_F^2 \leq 4\left(\frac{1+\lambda^2}{2}\right)^{t-s-1} L^2 p_s r_0^2 + \frac{1}{2304\eta C_T}(1+\eta\gamma)^{2(t-s-1)} p_s^2 r_0^2$$

$$+ \frac{L^2\eta^2(1+\eta\gamma)^{2(t-s-1)} L^2 p_s^2 r_0^2}{16 b_1 (t-s)^2} \tag{63}$$

because of the parameter setting. We should notice that here we also use the relation $\frac{\eta}{b_1\gamma} \leq O(1)$, which is always satisfied according to the setting of $b_1$. Based on Eq. (53) and Eq. (63), it is easy to check that $\zeta_t$ satisfies Eq. (40). Moreover, we have

$$\eta \sum_{\tau=s+1}^{t-1} (1+\eta\gamma)^{t-\tau-1}\|\zeta_\tau\|$$

$$\leq L\eta(1+\eta\gamma)^{t-s-1} p_s r_0 \left(8 \sum_{\tau=s+1}^{t-1}\left(\frac{4+\lambda^2}{5}\right)^{t-s-1} + \sum_{\tau=s+1}^{t-1}\frac{1}{\tau-s}\right) + \frac{1}{12}(1+\eta\gamma)^{t-s-1} p_s r_0$$

$$\leq L\eta\left(\frac{80}{1-\lambda} + \log(C_T)\right)(1+\eta\gamma)^{t-s-1} p_s r_0 + \frac{1}{12}(1+\eta\gamma)^{t-s-1} p_s r_0$$

$$\leq \frac{1}{6}(1+\eta\gamma)^{t-s-1} p_s r_0 \tag{64}$$

where the first inequality is derived by

$$\frac{1+\lambda^2}{2} \leq \left(\frac{3+\lambda^2}{4}\right)^2 \quad \text{and} \quad \frac{(3+\lambda^2)(1+\eta\gamma)}{4} \leq \frac{4+\lambda^2}{5} \tag{65}$$

Now combining Eq. (41), Eq. (51) and Eq. (64), we can reach the conclusion in Eq. (38) and finish the proof of the induction. Recall the assumption at the beginning, we have

$$\frac{1}{2}(1+\eta\gamma)^{C_T} p_s r_0 \leq w_{C_T} \leq 2\bar{d} \leq 6C_d \tag{66}$$

since $\|\bar{x}_t - \bar{x}'_t\| \le \|\bar{x}_t - \bar{x}_s\| + \|\bar{x}'_t - \bar{x}_s\|$. Eq. (66) implies that

$$C_T \le \frac{\log(12C_d/(p_s r_0))}{\log(1 + \eta\gamma)} < \frac{2\log(12nC_d/r_0)}{\eta\gamma} \tag{67}$$

which conflicts with the definition of $C_T$. Therefore, the proof of Lemma 6 is finished. $\qquad\square$

### C.7  Proof of Lemma 7

*Proof.* If node $i$ enters the escaping phase in iteration $s'$ before iteration $s$ and does not break it in iteration $s + C_T$, then for $s \le t \le s + C_T$, we have $\|x_t^{(i)} - x_s^{(i)}\| \le \|x_t^{(i)} - x_{s'}^{(i)}\| + \|x_s^{(i)} - x_{s'}^{(i)}\| \le 2C_d$. Therefore, there are at least $\frac{n}{10}$ worker nodes satisfying $\max_{s \le t \le s+C_T} \|x_t^{(i)} - x_s^{(i)}\| \le 2C_d$.

Suppose $\min eig(\nabla^2 f(\bar{x}_s)) \le -\epsilon_H$ and $\mathbf{e_1}$ is the corresponding eigenvector. Let $\mathcal{S}_i$ denote the region of the perturbation on node $i$ that PEDESTAL will terminate in iteration $s + C_T$, *i.e.*, $\frac{n}{10}$ workers will not break the escaping phase. Then by Lemma 6 we can conclude that there must exist one worker node such that the projection of $\mathcal{S}_i$ onto direction $\mathbf{e_1}$ is smaller than $r_0$. Since the perturbation $\xi_i$ is drawn from uniform distribution, the probability of $\xi_i \in \mathcal{S}_i$ can be bounded by

$$Pr(\xi_i \in \mathcal{S}_i) \le \frac{r_0 V(Ball(d-1,r))}{V(Ball(d,r))} \le \delta \tag{68}$$

where $V(\cdot)$ denotes the volume and $Ball(d,r)$ denotes the $d$-dimensional ball with radius $r$. The last inequality is achieved by the definition of $r_0$. Therefore, we can prove that $\bar{x}_s$ is a second-order stationary point with probability at least $1 - \delta$. $\qquad\square$

## D  Additional Theoretical Result

In this section we will provide some additional theoretical result of our PEDESTAL algorithm. First we will demonstrate the convergence analysis of the case $\epsilon_H < \sqrt{\epsilon}$, *i.e.*, $\alpha > 0.5$. Next, we will discuss the strategy of using fixed number of iterations in each descent and escaping phase, which motivates the design of PEDESTAL.

### D.1  Smaller Tolerance for Second-Order Optimality

When $\epsilon_H < \sqrt{\epsilon}$, the conclusions of previous Lemmas are still satisfied except Lemma 4. In this case, $C_d = C_2\eta C_T\epsilon^\mu$ where $\mu = 2\alpha > 1$. Parameter $C_d$ should be smaller than the original setting in Lemma 4, which results in more iterations to converge. Fortunately, the analysis of Lemma 4 can be adjusted and we can achieve Theorem 2. The proof is provided as follows.

*Proof.* The fourth term of $\mathcal{D}_t$ in Lemma 3 is derived by $\frac{\eta\beta\sigma^2}{b_1}$ and at this time we will set $b_1 \ge \epsilon^{-(2\mu-1-\theta)}$ so that the $\epsilon$ term is replaced by $\epsilon^\mu$. The last term of $\mathcal{D}_t$ can be written as

$$\sum_{t=0}^{T-1} \frac{7p_t(\eta^2 C_v^2 + r^2)}{4\eta} = \frac{1}{n} \sum_{(t,i)\in\mathcal{P}} \frac{7(\eta^2 C_v^2 + r^2)}{4\eta} \tag{69}$$

where $\mathcal{P}$ is the set of all pairs of $(t,i)$ such that node $i$ draws perturbation in iteration $t$. We can divide $\mathcal{P}$ into two parts. $\mathcal{P}_1$ contains all pairs of $(t,i)$ such that node $i$ breaks the escaping phase within $M$ iterations, where $M$ is an integer to be decided later. The rest part is denoted by $\mathcal{P}_2$.

For any $(t,i) \in \mathcal{P}_1$, suppose node $i$ breaks escaping phase in iteration $t + m$, where $m \le M$. Then node $i$ will never draw perturbation between iteration $t$ and iteration $t + M$. Mimic the steps of Eq. (26), by Cauchy-Schwartz inequality we can obtain

$$\sum_{\tau=t}^{t+m} \|x_{\tau+1}^{(i)} - x_\tau^{(i)}\|^2 \ge \frac{C_d^2}{M} \tag{70}$$

Let $M = \epsilon^{-2-2\theta+2\alpha}$. Then we have

$$\frac{(1-\lambda)^2}{512\eta} \sum_{\tau=t}^{t+m} \|x_{\tau+1}^{(i)} - x_\tau^{(i)}\|^2 \ge \frac{7(\eta^2 C_v^2 + r^2)}{4\eta} \tag{71}$$

and

$$\frac{(1-\lambda)^2}{512n\eta}\sum_{\tau=t}^{t+m}\sum_{i=1}^{n}\|x_{\tau+1}^{(i)}-x_{\tau}^{(i)}\|^2 \geq \frac{1}{n}\sum_{(t,i)\in\mathcal{P}_1}\frac{7(\eta^2C_v^2+r^2)}{4\eta} \tag{72}$$

by the parameter setting of $C_v$. On the other hand, if $(t,i) \in \mathcal{P}_2$, then node $i$ will not break the escaping phase in $M$ steps and hence the perturbation step will not execute, either. Therefore, we have estimation

$$\frac{1}{n}\sum_{(t,i)\in\mathcal{P}_2}\frac{7(\eta^2C_v^2+r^2)}{4\eta} \leq \sum_{t=0}^{T-1}\frac{7(\eta^2C_v^2+r^2)}{4M\eta} \tag{73}$$

With Eq. (72), Eq. (73) and the new setting of $b_1$, the descent in Lemma 3 can be improved to

$$\mathcal{D}_t = \frac{1}{16\eta}\omega_t + \frac{(1-\lambda)^2}{512n\eta}\sum_{i=1}^{n}\|x_{t+1}^{(i)}-x_t^{(i)}\|^2 + \frac{\eta}{2n}\sum_{i=1}^{n}\|y_t^{(i)}\|^2 - \frac{200\eta\epsilon^{2\mu}\sigma^2}{(1-\lambda)^2C_1^2} - \frac{7(\eta^2C_v^2+r^2)}{4M\eta}$$

When $\theta \geq 3\alpha - 2$, we have $\frac{\epsilon^2}{M} \leq \epsilon^{2\mu}$ and Lemma 4 still holds but the conclusion is changed to

$$\sum_{t\in\mathcal{I}}\mathcal{D}_t \geq |\mathcal{I}| \cdot \frac{(1-\lambda)^2C_2^2\eta\epsilon^{2\mu}}{10000}$$

In this case, PEDESTAL algorithm will terminate in $\tilde{O}(\epsilon^{-\theta-2\mu})$ iterations. In Lemma 6 and Lemma 7 we need the relations

$$\frac{\eta^2C_d^2C_T^3}{b_1} \leq O(1), \quad \frac{\eta}{b_1\epsilon_H} \leq O(1) \tag{74}$$

which implies $b_1 \geq \tilde{O}(\epsilon^{-\theta-\alpha})$. Therefore, we set $b_1 = \tilde{\Theta}(\epsilon^{-\max\{4\alpha-1-\theta,\theta+\alpha\}})$ with the condition $\theta \geq 3\alpha - 2$. When $\alpha \leq 1$, we set $\theta = \frac{3\alpha-1}{2}$, which satisfies $\theta \geq 3\alpha - 2$ and

$$4\alpha - 1 - \theta = \theta + \alpha = \frac{5\alpha-1}{2} \tag{75}$$

The gradient complexity in this case is

$$\tilde{O}(\epsilon^{-\frac{11\alpha-1}{2}} \cdot \epsilon^{-\frac{5\alpha-1}{2}}) = \tilde{O}(\epsilon^{-8\alpha+1}) \tag{76}$$

When $\alpha > 1$, we have $\theta = 3\alpha - 2$ and $b_1 = \tilde{\Theta}(\epsilon^{-(4\alpha-2)})$. The gradient complexity is

$$\tilde{O}(\epsilon^{-(7\alpha-2)} \cdot \epsilon^{-(4\alpha-2)}) = \tilde{O}(\epsilon^{-11\alpha+4}) \tag{77}$$

which finishes the proof of Theorem 2. □

Therefore, the gradient complexity over all cases of $\alpha$ can by written by

$$\tilde{O}(\epsilon^{-3} + \epsilon\epsilon_H^{-8} + \epsilon^4\epsilon_H^{-11}) \tag{78}$$

### D.2 Phases with Fixed Number of Iterations

If a decentralized stochastic perturbed gradient descent method adopt the strategy of fixed number of iterations in each phase, the gradient complexity in the descent phase should be at least $O(\epsilon^{-3})$ to ensure the first-order stationary point. But the total descent of a descent phase could be small because it is possible that it is stuck at a saddle point after only a few steps. Hence we need to consider the descent in the escaping phase. According to Lemma 3 and Lemma 4 we can see the descent of an escaping phase is $O(\frac{C_d^2}{\eta C_T})$. As the conditions $\eta C_d C_T \leq O(1)$ and $C_T = \tilde{O}(\frac{1}{\eta\epsilon_H})$ are required in Lemma 6, we can obtain that the total descent of an escaping phase is no larger than $\tilde{O}(\epsilon_H^3)$. In the classic setting of $\epsilon_H = \sqrt{\epsilon}$, the total descent of an escaping is upper bounded by $\tilde{O}(\epsilon^{1.5})$. Consequently, the total gradient complexity to achieve $(\epsilon, \sqrt{\epsilon})$-second-order stationary point is at least $\tilde{O}(\epsilon^{-4.5})$, which is worse than the result of our PEDESTAL.

# E   Auxiliary Lemmas

**Lemma 8.** *Let $0 < a < 1$. Then we have*

$$\sum_{\tau=1}^{t} \tau a^{\tau-1} = \frac{1-a^t}{(1-a)^2} - \frac{ta^t}{1-a}$$

**Lemma 9.** *Let $0 < a < 1$. When $t \geq \tilde{O}(\frac{1}{1-a})$, we have*

$$\sum_{\tau=1}^{t} \frac{\tau a^{\tau-1}}{(t+1-\tau)^2} \leq \frac{8}{t^2(1-a)^2}$$

*Proof.* When $\tau \leq \frac{t}{2}$, by Lemma 8 we have

$$\sum_{\tau \leq t/2} \frac{\tau a^{\tau-1}}{(t+1-\tau)^2} \leq \frac{4}{t^2(1-a)^2} \tag{79}$$

When $\tau > \frac{t}{2}$, we have

$$\sum_{\tau > t/2} \frac{\tau a^{\tau-1}}{(t+1-\tau)^2} \leq \sum_{\tau > t/2} \tau a^{\tau-1} \leq a^{t/2}\left(\frac{t}{2(1-a)} + \frac{1}{(1-a)^2}\right) \tag{80}$$

Therefore, we can reach the conclusion when $t \geq \tilde{O}(\frac{1}{1-a})$. $\qquad\square$

**Lemma 10.** *(Definition of Variance) For any random variable X, we have*

$$\mathbb{E}[X - \mathbb{E}X]^2 = \mathbb{E}X^2 - (\mathbb{E}X)^2$$

**Lemma 11.** *(Lemma D.1 in (Chen et al. [2022])) Let $\epsilon_{1:k} \in \mathbb{R}^d$ be a vector-valued martingale difference sequence with respect to $\mathcal{F}_k$, i.e., for each $k \in [K]$, $\mathbb{E}[\epsilon_k|\mathcal{F}_k] = 0$ and $\|\epsilon_k\| \leq B_k$, then with probability $1 - \delta$ we have*

$$\|\sum_{k=1}^{K} \epsilon_k\|^2 \leq 4\log(4/\delta) \sum_{k=1}^{K} B_k^2 \tag{81}$$

