# OpenReview forum: "Finding Local Minima Efficiently in Decentralized Optimization"
_NeurIPS.cc/2023/Conference — NeurIPS 2023 poster_

### Official Review · Reviewer_C5yK · 2023-06-29

**Soundness:** 3 good
**Presentation:** 2 fair
**Contribution:** 3 good
**Rating:** 6
**Confidence:** 3

**Summary:**

In "Finding Local Minima Efficiently in Decentralized Optimization" the authors propose a perturbed variance reduction stochastic optimization scheme that finds second order stationary points over a mesh network. The authors establish convergence results of the order \tilde(O)(\epsilon^{-3})  and verify empirically that their theoretical findings hold. Overall I think this work is nice, I follow with some criticism in the following sections.

**Strengths:**

1. Originality: To the reviewer's knowledge this seems to be the first paper in the (stochastic) decentralized set-up to establish second order guarantees with a given rate. Consequently, in this regard the paper's novelty seems to be sufficient.
2. Quality: The provided result seems to be technically sound.
3. Clarity: The presentation of the paper and the use of language can be improved. In many situations, I feel the authors' comments do not provide an intuition on the scheme's dependence on relevant quantities but provide an explanation based on what can be controlled or not in the proof. I provide concrete feedback on this in "Questions."
4. Significance: I think this work is significant given that it is the first to provide this type of guarantees in the decentralized set-up.

**Weaknesses:**

1. It should be clear from the main text which polylog factors the notation \tilde{O} is hiding. For this type of result, I would expect some scaling with the dimension (even if logarithmic) which should be highlighted at  some point.
2. Experimental set-up: The results in this work hold under the assumptions that the gradients and hessian are Lipschitz. The problem upon which the simulations are run fulfills neither of these assumptions. I understand that the numerical simulations confirm that the scheme converges even if the assumptions are violated, but there is currently no way of understanding wether the order of number of gradient evaluations that it takes to achieve a certain precision is close to tight (given that the simulations work with a more general class of functions than that covered by the theory).
3. Termination criteria: the scheme returns \bar{x}_{t-C_T} if there are at least n/10 nodes satisfying esc^i \geq  C_T. This seems to require global knowledge of the system. In other words, this does not seem implementable. As it is written, it seems to be the scheme does not terminate if this condition does not hold. The authors should clarify this. I see that in Section 3.2.3 the authors motivate the termination criteria via avoidance of the consensus error on the tracking variable (which is also not implementable), but this does not address how to implement the criteria.



**Questions:**

Questions/Suggestions/Comments:
1. There are paragraphs in which the language needs to be polished. See for example lines 45 to 48. I
2. In 2.2. the authors state that Perturbed SGD requires a complexity of O(epsilon^{-8}) to obtain a second order stationary point, and that this further hides a polynomial scaling with the dimension. My understanding is that the complexity is of order O(epsilon^{-4}) with logarithmic dependence with the dimension, see, e.g. Theorem 16 in https://arxiv.org/abs/1902.04811 (or 17 for the mini-batch).
3.  If my comment in 2 is correct, the way to justify the current work, as opposed to a decentralized extension of perturbed stochastic gradient descent (which would yield a much simpler scheme) is through the benefits of variance reduction.
4. The termination criteria is problematic. This is covered in detail under Weaknesses.
5. In my view the main technical contribution of this work should be to extend the "small stuck region" lemma to the decentralized case. In this sense, I would expect a similar intuition as that highlighted in https://arxiv.org/abs/1902.04811 with further complications due to the consensus error(s). Instead, the intuition provided in 3.2.4. is entirely based on the quantities that can or can not be controlled in the proof. I think having a geometric interpretation instead would make the paper more understandable.

---

> ### Author Rebuttal · Authors · 2023-08-02
>
> We appreciate the review of our paper and we will answer your questions as follows.
> 1. Thank you for the suggestion. In our final version we will go through the paper carefully and revise those grammar and language mistakes.
> 2. In our paper, we adopt the general bounded variance assumption on the stochastic gradient estimator. The baselines in this paper are also studied under this condition.  However, in [1] a stronger assumption on the tail of data distribution is used (see Assumption B in [1]). We will list and discuss this result in our final version.
> 3. The limitation of [1] has been mentioned in the answer to Question 2, which requires a stronger assumption on the data distribution. Under such assumption, we think it is feasible to extend the simpler scheme in [1] to the decentralized setting since the perturbation is drawn from Gaussian distribution (the mixture of Gaussian distributions is also Gaussian distribution).
> 4. Thank you for the question. The termination condition is used to guarantee a second-order stationary point theoretically, which can be satisfied with high probability. In practice, we can set a maximum iteration (as we have done in our experiment). Generally we need to evaluate the model after certain epochs to see if the training process is running properly and we can save a checkpoint when we find a better evaluation result. The theoretical analysis ensures that an optimal solution can be visited if the number of iterations is as large as $O(\epsilon^{-3})$. Moreover, if we need the termination condition to avoid unnecessary iterations, we can run a separated process to collect the information when calculating gradients and transmitting models among neighborhoods, since obtaining the global knowledge of bool values (whether $esc^{(i)} \ge C_T$) is relatively less expensive.
> 5. In addition to the shape of stuck region, another important question is why there exists a stuck region. Suppose $x_0$ is a saddle point and $x$ is in the neighborhood of $x_0$.  If $x – x_0$ is in the eigenvector direction corresponding to a negative eigenvalue of $\nabla^2 f(x_0)$, then the negative gradient is a descent direction and $x$ moves away from $x_0$. Consequently, the stuck region is proven to be contained in a ``thin pancake” as shown in figure 1 in [1]. However, if the consensus error is not under control, it can drive $x$ away from $x_0$ or push $x$ toward $x_0$, no matter what $\nabla f(x)$ is. In this manner, the stuck region cannot be estimated. Besides, we would like to point out that the theoretical result of the termination section which categorizes all iterations into three types is also an important technical contribution of this paper. It forms the complete theorem together with the generalized small stuck region lemma. The small stuck region lemma provides guarantees for the case that the escaping phase is not broken while the termination section deals with the case that the escaping phase is broken.
>
> [1] On Nonconvex Optimization for Machine Learning: Gradients, Stochasticity, and Saddle Points, https://arxiv.org/abs/1902.04811

---

> > ### Comment · Reviewer_C5yK · 2023-08-14
> >
> > Thank you for your answers. I would suggest that the authors highlight the answers to 3 and 2 in the paper as it answers the question "why do I really need variance reduction in this set-up". I am happy with how the authors addressed my concerns (except the numerical simulation, under weaknesses). I have increased my score.

---

### Official Review · Reviewer_aGCa · 2023-07-01

**Soundness:** 2 fair
**Presentation:** 3 good
**Contribution:** 3 good
**Rating:** 5
**Confidence:** 4

**Summary:**

This paper presents a novel decentralized algorithm for escaping saddle points for non-convex problems. The authors focus primarily on the second-order optimality. Specifically, they propose a gradient-based stochastic optimization algorithm PEDESTAL with a novel convergence analysis framework. They provide the theoretical guarantees with the best available gradient complexity to achieve the second-order stationary point. To validate the proposed algorithm, the authors present two empirical results with the synthetic and real-world datasets.

**Strengths:**

I think the investigated topic is very interesting. The second-order optimality of decentralized stochastic optimization algorithm has not sufficiently been investigated. Most of existing works focused only on finding the first-order stationary point. It is exciting to see a fresh work that investigates the second-order optimality. The paper is easy to follow, though there are some typos that need to be corrected.

**Weaknesses:**

In order to make the paper more technically solid and sound, the authors may have to pay attention to the following aspects.

1. In the algorithm framework, the authors use $n/10$ to terminate the algorithm. Why? Why not $n/5$ or $n/2$? Any specific criterion to determine such a certain number of nodes? Otherwise, it would be odd to just have this condition.

2. The authors mentioned that broadcasting or aggregation is not required. But what about step 8 or 15? Did I misunderstand something here?

3. The experimental results are not promising. More complex datasets and models are required to validate the proposed algorithm. In the discussion, the authors also said that all baselines were stuck at the saddle point and couldn’t escape it efficiently. How to guarantee the point these baselines were stuck at is a saddle point? Not just a bad local minimum? I think the authors should be careful about their claim. It would be better to see a toy example problem that includes obvious saddle points and local minima. Then the authors show exactly how the proposed algorithm and baselines perform.

4. Some confusion about the table. The authors showed that the PEDESTAL has the larger gradient complexity than GT-HSGD. However, in terms of empirical plots, the PEDESTAL had faster and better convergence rate. Why? Moreover, the classic setting for the PEDESTAL-S is missing.


**Questions:**

1. How to quantitatively decide the termination criterion for the PEDESTAL algorithm?
2. How does the proposed algorithm perform in a non-IID data distribution setting?
3. How to distinguish the bad local minima from the saddle points in terms of the results presented in the paper?

**Limitations:**

Please include the discussion of the limitations for the proposed algorithms in the paper.

---

> ### Author Rebuttal · Authors · 2023-08-02
>
> Thank you for your review and we will address your concerns as follows.
> 1. We can use any constant $\alpha \in (0, 1)$ in the terminate condition. In our paper we set $\alpha = 1/10$ to simplify the computation in the convergence analysis. $\alpha$ can also be set to $1/5$ or $1/2$, but some coefficients in the proof should be changed correspondingly. We will clarify it in our final version.
> 2. Step 8 and step 15 only require communications among neighborhoods, without broadcast over the whole network. If node i and node j are not connected in the network, $w_{ij}$ is 0.
> 3. Both two experiments in this paper are non-spurious local minimum problems, which means all local minima are global minima (see [1] and [2]). If the loss function value does not meet the global minimum, then the reached point is not a local minimum. Therefore, it is guaranteed that baselines are stuck at saddle points, not just bad local minima. We will add the explanation in the description of each task instead of putting it in the front of Section 5.
> 4. First we need to point out that PEDESTAL matches the gradient complexity of GT-HSGD in the classic setting of $\epsilon_H = \sqrt{\epsilon}$, ignoring logarithm terms. Moreover, the gradient complexity of GT-HSGD only guarantees the first-order optimality, which include both local minimum and saddle point. In contrast, the gradient complexity of PEDESTAL guarantees second-order optimality. In the experiments, PEDESTAL shows advantageous performance because PEDESTAL is better at escaping saddle points. PEDESTAL-S is a special case of PEDESTAL that the batchsize is fixed as $O(1)$ and the analysis of PEDESTAL-S only works for $\epsilon_H \ge \epsilon^{0.2}$, which does not cover the classic setting.
>
> [1] No spurious local minima in nonconvex low rank problems: A unified geometric analysis, ICML 2017
>
> [2] Matrix completion has no spurious local minimum, NeurIPS 2016

---

> > ### Comment · Reviewer_aGCa · 2023-08-16
> > **Response to the Rebuttal**
> >
> > Thanks the authors for the clarification. I think the rebuttal addressed most of my concerns, though the empirical results still look a bit weak. I hope the authors will make all corresponding changes based on the comments from all reviewers.

---

### Official Review · Reviewer_D1UK · 2023-07-04

**Soundness:** 3 good
**Presentation:** 3 good
**Contribution:** 3 good
**Rating:** 7
**Confidence:** 4

**Summary:**

This paper proposes PEDESTAL, decentralized optimization algorithm that can find second-order stationary points efficiently, and provides extensive numerical experiments to support theoretical analysis. PEDESTAL outperforms other existing decentralized second-order algorithms as it requires a smaller batch size and less iterations to converge.

**Strengths:**

1. The proposed algorithm is simple but has a stronger performance guarantee than existing algorithms. The proof requires only standard assumptions.
2. In addition to having superior convergence guarantees, the proposed algorithm outperforms other algorithms in numerical experiments.

**Weaknesses:**

1. The stopping criteria requires global synchronization. Even though this algorithm only requires to send an integer over the network, it is still required to sync every iteration to decided whether to stop.
2. Rates don't include the eigen gap. I suggest the authors to explicitly show this terms, which gives us a better idea of the algorithm's efficiency.
3. Notations are not clearly defined in the paper. I was not able to find the definition of $\tilde O(\cdot)$.
4. Some minor typesetting flaws. Page 7 line 276 - 279, should be $x_1$ instead of $x1$.

**Questions:**

I have a question about numerical experiments. Are all

---

> ### Author Rebuttal · Authors · 2023-08-06
>
> We appreciate the recognition of our contribution and we will address you concerns as follows.
> 1. To avoid synchronization, an alternative solution of the termination condition is to set an upper bound of the number of iterations. We can evaluate the training model at certain frequency and save the status if it achieves the best result so far. The theoretical analysis guarantees that the optimal solution will be visited, though the algorithm probably does not stop at this point.
> 2. According to current proof, the iteration complexity on eigen gap is $O(\frac{1}{(1 - \lambda)^2})$ and the total complexity on eigen gap is $O(\frac{1}{(1 - \lambda)^4})$. However, in our work we focus on the second-order optimality of decentralized optimization and hence some unnecessary relaxations on eigen gap are probably applied because of simplicity. Therefore, this result can probably be improved and we will work on it to add it in our final version.
> 3. Thank you for pointing out missing definitions and typos. In our final version we will introduce all notations used in this paper at the end of Section 1. We will also revise our paper to correct typos and grammar mistakes.

---

> > ### Comment · Reviewer_D1UK · 2023-08-12
> > **Response**
> >
> > Thanks for the clarification.

---

### Official Review · Reviewer_eGsh · 2023-07-07

**Soundness:** 3 good
**Presentation:** 3 good
**Contribution:** 2 fair
**Rating:** 6
**Confidence:** 2

**Summary:**

This paper proposed a gradient-based decentralized stochastic algorithm in decentralized optimization. The second-order optimality is proved with non-asymptotic analysis.

**Strengths:**

The paper is well-written and easy to follow. The mathematical proof seems rigorous.

**Weaknesses:**

The numerical studies can be improved.

**Questions:**

I have some concerns on the numerical studies part. In the tasks studied, the performance proposed method seems to be too strong as compared to other baselines. All methods except the proposed one stuck at the saddle point. Maybe the author can consider some common machine learning jobs like image classification and do some comparsions.

**Limitations:**

See questions.

---

> ### Author Rebuttal · Authors · 2023-08-05
>
> Thank you for your review and suggestion and we will address your concerns. In this paper the two experiments we conducted are both non-spurious local minimum problems according to [1] and [2], which means all local minima are global minima. Therefore, we can conclude when an algorithm is stuck at saddle point by comparing the loss function value and thus evaluate the capability to escape saddle points. Otherwise, in some general problems such as image classification we cannot distinguish a saddle point or a bad local minimum.
>
> [1] No spurious local minima in nonconvex low rank problems: A unified geometric analysis, ICML 2017
>
> [2] Matrix completion has no spurious local minimum, NeurIPS 2016

---

> > ### Comment · Reviewer_eGsh · 2023-08-14
> > **Response**
> >
> > Thanks for the response. I have no further questions and will increase the score to 6.

---

### Official Review · Reviewer_tsG4 · 2023-07-09

**Soundness:** 3 good
**Presentation:** 4 excellent
**Contribution:** 3 good
**Rating:** 7
**Confidence:** 4

**Summary:**

This work proposes PEDESTAL, the first decentralized algorithm for the stochastic non-convex optimization framework that escapes saddle points efficiently. Theoretical guarantees are provided indicating that PEDESTAL finds second order stationary points in $\tilde{\mathcal{O}} (\epsilon^{-3} + \epsilon_H^{-5})$. In the classical setting where $\epsilon_H = \sqrt{\epsilon}$ this translates to $\tilde{\mathcal{O}$ which matches the state of the art results for finding a first order stationary point. To achieve this aforementioned result the authors utilize perturbed gradient decent (from the centralized regime) and STORM to accelerate convergence. Importantly, the ideas of perturbed gradient descent (i.e. the injection of noise followed by a phase that verifies whether a second order stationary point is reached) are utilized in a distributed manner thus improving from prior work and requiring no coordination between the nodes. Additionally, numerical results on matrix sensing and matrix factorization tasks are provided supporting the theoretical findings.

**Strengths:**

- The paper is $\textbf{very well written}$ and easy to follow for the most part (more details on this in the weaknesses section.)

- This paper studies an $\textbf{interesting problem}$ in the field of non-convex stochastic optimization i.e. finding efficiently second order stationary points in a decentralized manner.

- $\textbf{Novel techniques}$. The paper combines ideas from prior works (perturbed gradient descent, perturbed gradient tracking, STORM estimator) in a non-trivial manner. The theoretical contribution is significant since the authors utilize ideas of perturbed gradient descent in a decentralized manner without the need for any coordination among the participating nodes. I believe that this is a novel and interesting technique that can be used by more decentralized methods in the future to escape efficiently from saddles.

- $\textbf{Impactful results}$. The resulting algorithm and analysis are impactful deriving the first decentralized algorithm for the stochastic non-convex optimization framework that escapes saddle points efficiently with convergence guarantee $\tilde{\mathcal{O}$; this is indeed a competitive rate matching the best known rates for finding first order stationary points.

- $\textbf{Numerical results}$ that support the theoretical findings are provided showcasing that PEDESTAL efficiently escapes saddle points.

**Weaknesses:**

- Although the paper has very nice structure and is well-written, I believe that the presentation of $\textbf{section $3.2.4$ could be improved}$. Arguably one of the most interesting ideas of this work is the implementation of the descent and escape phases in a fully distributed fashion without the need for any coordination. I believe that conveying the intuition of how this is achieved in PEDESTAL is of crucial importance that needs to be emphasized. Although, the authors have attempted to do that in section $3.2.4$ I think that this part of the text should be revised explaining the underlining ideas more clearly and in more detail. Similarly, very little intuition or discussion has been for the proof of the important Lemma $6$ in the appendix.

- I would like the authors to clarify how the algorithm terminates i.e. how the clients know that $n/10$ of them satisfy the $\textbf{termination condition}$.

- The bound provided in Theorem $1$ holds for a $\textbf{specific range of}$   $\boldsymbol{\alpha \leq 0.5}$. Although it is true that $\alpha =0.5$ covers the most important (classic) setting, this limitation should be clearly stated in the main body of the paper (a footnote could suffice).

- Typos:
Line 170 "REDESTAL" change to "PEDESTAL".
Line 171 change "two phase" to "two phases".
Line 204 "In Pullback...as SPIDER" needs rephrasing.
Line 214 "If the...is used" needs rephrasing.


**Questions:**

See the weaknesses section.

**Limitations:**

The authors adequately addressed the limitations of their work.

---

> ### Author Rebuttal · Authors · 2023-08-05
>
> We appreciate the recognition of the contribution of our work and will address you concerns as follows.
> 1. Thank you for your suggestion. We will add more details to convey the intuition of Section 3.2.4 and Lemma 6 more clearly. We will add a figure to show the relations of moving distances, the challenge and our solution.
> 2. We have two strategies to implement the termination condition in practice. The first one is to run a separated process to obtain the number of nodes satisfying the stopping condition at the same time of gradient calculation, communication and evaluation since transmitting bool value is relatively less expensive. The second solution is to set up the number of iterations in advance. The model is evaluated after certain number of iterations such that we can check if the training process is running properly and plot figures based on these results. If the evaluated model achieves the current best result, we will save this checkpoint. Our theorem guarantees that the optimal solution must be visited if the number of iterations is $O(\epsilon^{-3})$.
> 3. Thank you for the suggestion. We will add a note of the range of $\alpha$ in Theorem 1. In the appendix, we provide another Theorem 2 that proves the case of $\alpha > 0.5$ using different parameter settings. In our final version, we will put both Theorem 1 and 2 in the main manuscript.
> 4. Thank you for pointing out the grammar errors and typos in our paper. We will revise and polish the language in our final version.

---

> > ### Comment · Reviewer_tsG4 · 2023-08-13
> > **Response to the Rebuttal**
> >
> >
> > I thank the authors for addressing my concerns. I would like, however to ask for a few more details on their response?
> >
> > -Specifically, could you please include the updated version which explains in details the intuition of Section $3.2.4$ as well as the intuition of Lemma $6$?
> >
> > -Furthermore, it is my understanding that the convergence rate of PEDESTAL has a logarithmic dependence on the dimension $d$. Please let me know if this is indeed the case.

---

> > > ### Author Response · Authors · 2023-08-16
> > >
> > > Thank you for the response. In our final version, we will add a figure in section 3.2.4 to show the relation between total moving distance and averaged moving distance. In this figure we will plot the iteration point $x_0, x_1, \cdots, x_t$ and the segment from $x_{i}$ to $x_{i+1}$ represents the update. Suppose the candidate point is $x_s$, $0<s<t$ (the start point of the small stuck region lemma). In our decentralized algorithm, some node can enter the escaping phase before iteration $s$ and we can assume it enters escaping phase at $x_0$. The averaged moving distance used in previous works is $\frac{1}{r} \sum_{i=0}^{r-1} || x_{i+1} – x_i ||^2 \le C_1$ for $r = 0, \cdots, t-1$, which is the average length of all segments. Since the sum starts from $x_0$, the averaged distance $\frac{1}{t-s} \sum_{i=s}^{t-1} || x_{i+1} – x_i ||^2$ can be as large as $\frac{t C_1}{t - s}$ if $x_1, \cdots, x_s$ are close to $x_0$, which cannot be bounded if $s$ is large. When we use total moving distance as the criterion (the distance between $x_i$ and $x_0$), we have $|| x_s – x_0 || \le C_2$ and $|| x_t – x_0 || \le C_2$. Then we must have $|| x_t – x_s || \le 2 C_2$, which can be used for estimation.
> > >
> > > The intuition of Lemma 6 is to prove that starting from the two statuses $X_s$ and $X_s’$ (defined in Lemma 6) and applying the algorithm with same random samples, the distance of $w_t = \bar{x}_t - \bar{x}_t’$ will be large enough after $t - s$ iterations. In the proof, we suppose the conclusion of Lemma 6 is not true and finally find the conflict. In specific, we use mathematical induction to prove that $|| w_t || \ge \frac{1}{2} (1 + \eta \gamma)^{t-s} p_s r_0$, which conflicts the condition $|| w_t || \le 6 C_d$.
> > >
> > > Besides, it is true that the convergence rate has a logarithmic dependence on the dimension $d$. Thank you for the question and we will add this comment in our final version to avoid reader’s confusion.

---

> > > > ### Comment · Reviewer_tsG4 · 2023-08-18
> > > > **Followup Response**
> > > >
> > > > Thank you for clarifying these points.
> > > > I find the rebuttal satisfactory and I am inclined to keep my score unchanged.

---

### Decision · Program_Chairs · 2023-09-21

**Decision:**

Accept (poster)

**Comment:**

The paper is the first of its kind to establish second-order guarantees with a given rate in a decentralized, stochastic setup. The technical soundness of the results is affirmed.  One weakness is that the experimental setup violates the paper's own assumptions about Lipschitz continuity for gradients and Hessians. Also, the experimental results are not promising. Overall, given its strong theoretical result, the value of this work outweighs its weakness.